# Source apportionment of particle number size distribution at the street canyon and urban background sites

Sami D. Harni[1*], Minna Aurela[1], Sanna Saarikoski[1], Jarkko V. Niemi[2], Harri Portin[2], Hanna Manninen[2], Ville Leinonen[3], Pasi Aalto[4], Phil K. Hopke[5], Tuukka Petäjä[4], Topi Rönkkö[6], Hilkka Timonen[1]

[1] Atmospheric Composition Research, Finnish Meteorological Institute, Helsinki, Finland
[2] Helsinki Region Environmental Services Authority (HSY), Helsinki, Finland
[3] Faculty of Science, Forestry and Technology, Department of Technical Physics, University of Eastern Finland, Finland
[4] Institute for Atmospheric and Earth System Research (INAR) / Physics, Faculty of Science, University of Helsinki, Finland
[5] Department of Public Health Sciences, University of Rochester School of Medicine and Dentistry, Rochester, NY 14642, USA
[6] Aerosol Physics Laboratory, Tampere University, Tampere, Finland

Corresponding author: Sami D. Harni, sami.harni@fmi.fi

**Abstract.** Particle size is one of the key factors influencing how aerosol particles affect their climate and health effects. Therefore, a better understanding of particle size distributions from various sources is crucial. In urban environments, aerosols are produced in a large number of varying processes and conditions. This study intended to develop the source apportionment of urban aerosols by utilising a novel approach to positive matrix factorization (PMF). The particle source profiles were detected in particle number size distribution data measured simultaneously in a street canyon and at a nearby urban background station between February 2015 and June 2019 in Helsinki, southern Finland. The novelty of the method is combining the data from both sites and finding profiles for the unified data. Five aerosol sources were found. Four of them were detected at both of the stations: slightly aged traffic (TRA2), secondary combustion aerosol (SCA), secondary aerosol (SecA), and long-range transported aerosol (LRT). One of the sources, fresh traffic (TRA1) was only detected at a street canyon. The factors were identified based on available auxiliary data. Additionally, the trends of the found factors were studied, and statistically significant decreasing trends were found for TRA1 and SecA. A statistically significant increasing trend was found for TRA2. This work implies that traffic-related aerosols remain important in urban environments and that aerosol sources can be detected by using only particle number size distribution data as input in the PMF method.

## 1 Introduction

Urban aerosol is a complex mixture of particles of various sizes and compositions originating from multiple anthropogenic and natural sources, including sea salt, fuel combustion (e.g., in thermal power generation, incineration, domestic heating, combustion engines), road, tire, and brake wear, dust, pollen, volcanic ash, forest fires, and industry (Almeida et al., 2006; Guerreiro et al., 2015; Karanasiou et al., 2009). Of these anthropogenic sources are predominant in urban areas (Guerreiro et al., 2015). The negative health effects related to particulate matter (PM) pollution ($PM_{2.5}$ and $PM_{10}$) are commonly accepted

and well-documented (i.e., Koenig, 2000; J. Wu et al., 2017), also leading to indirect financial consequences through increased mortality and treatment of respiratory and cardiovascular diseases (Johnston et al., 2021). The recent WHO good practice statement encourages the systematic measurement of particle number concentration (PNC) $\geq$ 10 nm, emphasizing the significance of PNC in addition to PM mass (WHO, 2021).

Source apportionment of aerosols can be done in multiple ways. Commonly used source apportionment techniques in atmospheric sciences include k-means cluster analysis, principal component analysis (PCA), and receptor modelling methods. In this work, a receptor modelling method called positive matrix factorization (PMF) was used. PMF is a mathematical multi-derivative method developed by Paatero, (1997) that can be performed for many types of data, and it is the most widely used and established source apportion method for atmospheric aerosol particle data currently (Hopke et al, 2020; Hopke et al., 2022; Yang et al, 2020). The decision to use PMF was made because PMF is a well-established source apportionment method in environmental sciences, and there was suitable software available. Additionally, as PMF is a factor analysis method, it is fundamentally suitable to this kind of study, as it assumes that the observed data is a combination of latent underlying factors. In contrast, PCA, for example, attempts to linearly combine the underlying variables to reduce the size of the data.  PMF has been used for chemical composition data (e.g. Li et al. 2003; Makkonen et al., 2023), mass spectra (e.g. (Oduber et al., 2021; Teinilä et al., 2022), particle number size distribution (PNSD) (Krecl et al., 2008; Zhou et al., 2005) and in combined matrixes with PNSD and auxiliary data (Rivas et al., 2020). However, conducting source apportionment solely based on PNSD data and using auxiliary data only to verify the sources seems to have some challenges as the source profiles might be mixed with multiple sources (Zhou et al., 2005, Jollife & Cadima, 2016, Krecl et al., 2008). This makes interpreting of results using auxiliary data more difficult. To improve the separation between sources when using only PNSD data as input to PMF, PNSD data from two sites is combined into one data file in this study.

Urban aerosol size distributions have been studied in a comprehensive review of urban aerosols consisting of approximately 200 articles, including 114 cities in 43 countries (Wu & Boor, 2021). They stated that in urban environments, the majority of particles are in the size range of 10-100 nm, and the PNC decrease approximately by a factor of 100 when the particle size increases from 100 nm to 1000 nm. In particular, PMF especially has been applied to size distribution data in numerous studies in urban, suburban, urban background, or residential locations in Asia, Australia, the Middle East, Europe, and the USA. (Al-Dabbous et al., 2015; Dai et al., 2021; Friend et al., 2012; Gu et al., 2011; Harrison et al., 2011; Kasumba et al., 2009; Kim et al, 2004; Krecl et al., 2008; Leoni et al., 2018; Liu, et al., 2017; Ogulei et al., 2007; Pokorná, et al., 2020; Rivas et al., 2020; Squizzato et al., 2019; Thimmaiah et al., 2009; Vu et al., 2016; Wang, et al., 2013; Yue et al., 2008; Zong et al., 2019). Only one of the studies used data from Helsinki (Rivas et al., 2020). In this study, particle PNSD was investigated in urban background (UB) and street canyon (SC) sites in Helsinki, southern Finland. The simultaneous data from these two sites have been analysed in previous studies. Okuljar et al. (2021) investigated the relative contribution of traffic and atmospheric new particle formation to the concentration of sub-3 nm particles. They utilized PNC data between 1-800 nm and auxiliary data from the stations. They found that the particle concentrations in the SC were higher over the whole size range. Additionally, they associated particles in the size range of 1-25 nm with local sources at the UB and found particles in the size range of 1-

100 nm to have a dominant contribution from local sources in the SC. Rivas et al., (2020) used data from both sites in a study that applied PMF on PNSD data across four European cities. They identified five factors for both stations: nucleation, fresh traffic, urban background, biogenic, and secondary.

Earlier studies conducted in the Helsinki metropolitan area have shown that PNSDs vary based on the dominant source. Nucleation-produced particles are the smallest, with mode particle sizes of 7-11 nm; traffic-influenced emissions have varying mode particle sizes of 10-75 nm, with the smaller particles produced by nucleation and larger particles from (Harni et al., 2023; Pirjola et al., 2017; Rivas et al., 2020). Woodburning has been shown to produce slightly larger particles at a mode particle size of 46 nm and to have a wide particle size distribution (Harni et al., 2023; Pirjola et al., 2017). Biogenic emissions have been shown to produce particles with mode sizes between 69 and 100 nm (Harni et al., 2023; Rivas et al., 2020).

This study was intended to improve the understanding of urban aerosol sources by applying statistical source apportionment methods such as PMF (EPA PMF 5) to long-term size distribution data. The factors were identified based on the diurnal cycles of PMF factors, available supporting data, including gases ($NO_x$, $CO_2$, $O_3$), and particle chemistry. The data used in this study was measured at two sites, an SC and UB from February 2015 to June 2019. These datasets, comprising more than 4 years also allowed indicative investigation of the trends in PNSDs and a discussion of the reasons for the changes observed in PNSDs. The PMF has been applied to the data from these two sites earlier in a study by Rivas et al., (2020). However, they used data from January 2007 to December 2016 whereas data in this study is from January 2015 to June 2019. Additionally, Rivas et al., (2020) used $NO_2$, NO, $SO_2$, CO, and $O_3$ data in addition to PNSD data in the PMF input files to separate the sources. In contrast, in this study only the PNSD data was included in the PMF input files and the results were later compared to the other measurement data. In addition to this, the novelty of this study arises from how the data was handled from the two nearby sites with strongly overlapping aerosol sources by adding the data from the two sites to the same data matrix horizontally as columns instead of doing two separate PMF analyses.

## 2 Experimental

### 2.1 Measurement sites

The data used in this study was measured at two atmospheric measurement stations. The first measurement station is an SC site at Mäkelänkatu in Helsinki Finland (60° 11' 47.53" N, 24° 57' 6.41" E), and it is governed by the Helsinki Region Environmental Services Authority. The measurement site is situated beside one of the busiest main roads in Helsinki and is heavily influenced by traffic emissions. The SC measurement site is described in detail by Barreira et al. (2021). The second station, the UB station, is located at Kumpula Helsinki SMEAR III ( 60° 12' 10.41" N, 24° 57' 40.53" E). The site is situated near a park area more than 100 m from the nearest busy road Järvi et al. (2009) describes the SMEAR III station in detail. The distance between the SC and UB stations is approximately 900 m.

## 2.2 Instruments

Particle PNSD data used in this work was measured between February 13, 2015 and June 5, 2019. The instruments used in the measurements are listed in Table 1. At the SC site, PNSDs were measured with a differential mobility particle sizer (DMPS) consisting of a condensation particle counter (CPC, A 20, Airmodus, Helsinki, Finland) and a Vienna-type differential mobility analyser (DMA). At the UB site, PNSDs were measured with a twin differential mobility particle sizer (Twin-DMPS). Hoppel (1978) describes the working principle of DMA and response functions in detail. The size spectra of DMPS at the SC site were measured in 26-size bins with particle sizes ranging from 6 to 800 nm and a time resolution of approx. 8 min 40 s to 9 min 5 s. The DMPS at the UB site measured particles in 50-size bins with particle sizes of 3-794 nm with a time resolution of approx.. 9 min 50 s to 10 min 5 s. Both of the DMPS systems were made by the University of Helsinki and approved by European Center for Aerosol Calibration and Characterization. Both of the systems had dryers in the inlet lines to keep the relative humidity (RH) below 40%. The DMPS charger had difficulties charging the three smallest particle size bins (6.0, 7.3, and 9.0 nm) on the SC site; therefore, particles smaller than 10 nm were excluded from the analysis for both sites. Both DMPS systems participated in an intercomparison with a reference instrument from Leibniz Institute for Tropospheric Research (TROPOS) SMPS in the UB station between June 11 and 14, 2021, and demonstrated comparable results. Non-refractory $PM_1$ (organics, sulphate, nitrate, ammonium, chloride) was measured with an aerosol chemical speciation monitor (Q-ACSM, Aerodyne Research Inc., [Ng et al., 2011]) at the SC site. A mass-based Q-ACSM calibration was performed using dried size-selected (300 nm of mobility diameter) ammonium nitrate and ammonium sulphate aerosol particles. The effective nitrate-response factor (RFNO3) relative ionization efficiencies of sulphate and ammonium (RIENH4) and relative ionization efficiency of sulphate (RIESO4) were determined, and analyte signals were converted into nitrate-equivalent mass concentrations. IE($NO_3$) varied over the years and the final correction of the NRPM1 mass was done against the mass concentrations derived from DMPS data as described by Barreira et al., (2021). The RIE for $SO_4$ varied from 0.51-0.61 and for $NH_4$ 3.8-5.32. An effusive source of naphthalene, located in the detection region, was used as a reference for m/z and ion transmission calibrations. A Nafion dryer was installed prior to the instrument inlet so that the RH of the sample flow was maintained below 40%. A chemical composition-dependent collection efficiency was used having been calculated according to Middlebrook et al. (2012), with the exception that a collection efficiency of 0.45 was used for samples when ammonium was below the detection limit. More information can be found in Barreira et al. (2021).

$PM_{2.5}$ and $PM_{10}$ mass concentrations were measured with a Tapered Element Oscillating Microbalance (TEOM, model 1405). The black carbon (BC) concentrations at the SC were measured using an Aethalometer (AE33, Magee Scientific). $NO_x$ and $O_3$ concentrations were measured with Horiba APNA 370 and Horiba APOA-370. At the SC, $CO_2$ concentrations were measured with LICOR model LI-7000 and CO with Horiba APMA-360. At the UB, $NO_X$ and $O_3$ were measured with Thermo Environmental Instruments 42S and 49. $SO_2$ and CO were measured with Horiba APMA 370 and Horiba, APSA 360.

**Table 1: The list of instruments used in the measurements.**

| Instrument | Station | Measured variable |
|---|---|---|
| DMPS (Airmodus CPC A20 with Vienna type DMA) | SC | PNSD |
| DMPS (Twin-DMPS, Aalto et al. 2001) | UB | PNSD |
| Q-ACSM (Aerodyne Research) | SC | Non-refractory $PM_1$ |
| TEOM (model 1405, Thermo Scientific) | SC | $PM_{10}$ and $PM_{2.5}$ |
| Aethalometer (AE33, Magee Scientific) | SC | BC |
| Ambient $NO_x$ Monitor (APNA-370, Horiba) | SC | $NO_x$ |
| Ambient Ozone Monitor (APOA-370, Horiba) | SC | $O_3$ |
| LICOR (model LI-7000) | SC | $CO_2$ |
| Ambient Carbon Monoxide Monitor (APMA-360, Horiba) | SC | CO |
| NO-$NO_2$-$NO_x$ Analyzer (42C, Thermo Environmental Instruments) | UB | $NO_x$ |
| $O_3$ Analyzer (model 49C, Thermo Environmental Instruments) | UB | $O_3$ |
| Ambient Carbon Monoxide Monitor (APMA- 370, Horiba) | UB | CO |
| Ambient Sulfur Dioxide Analyzer (APSA -370, Horiba) | UB | $SO_2$ |

## 2.3 Meteorology

Helsinki is a northern city with four seasons. The total radiance (Itot) and RH were measured beside the UB at the Helsinki Kumpula weather station (60.203071N, 24.961305E, 24 m asl). The temperature (T) used in this study was measured at the Helsinki Kaisaniemi weather station (60.17523N, 24.94459E, 3 m asl), situated 2.4 km south of the SC and 3.2 km from the UB; the T data measured at the Helsinki Kumpula weather station had a large gap in late 2017, missing several months of data.

The monthly average RH, T, and I$_{tot}$ are presented in Fig. 1. The RH reached maximum values during early winter (November-January), and the lowest values were measured during late spring (May). T and I$_{tot}$ reached maximum values during the summer months (June-August). The I$_{tot}$ reached the maximum slightly earlier (June-July) than the T (July-August). The highest monthly average T was measured during July 2018 (21.3 °C), and the lowest T was in January 2016 (-9.2 °C, Fig. 1c). In this paper, meteorological data was used in the interpretation of the results.

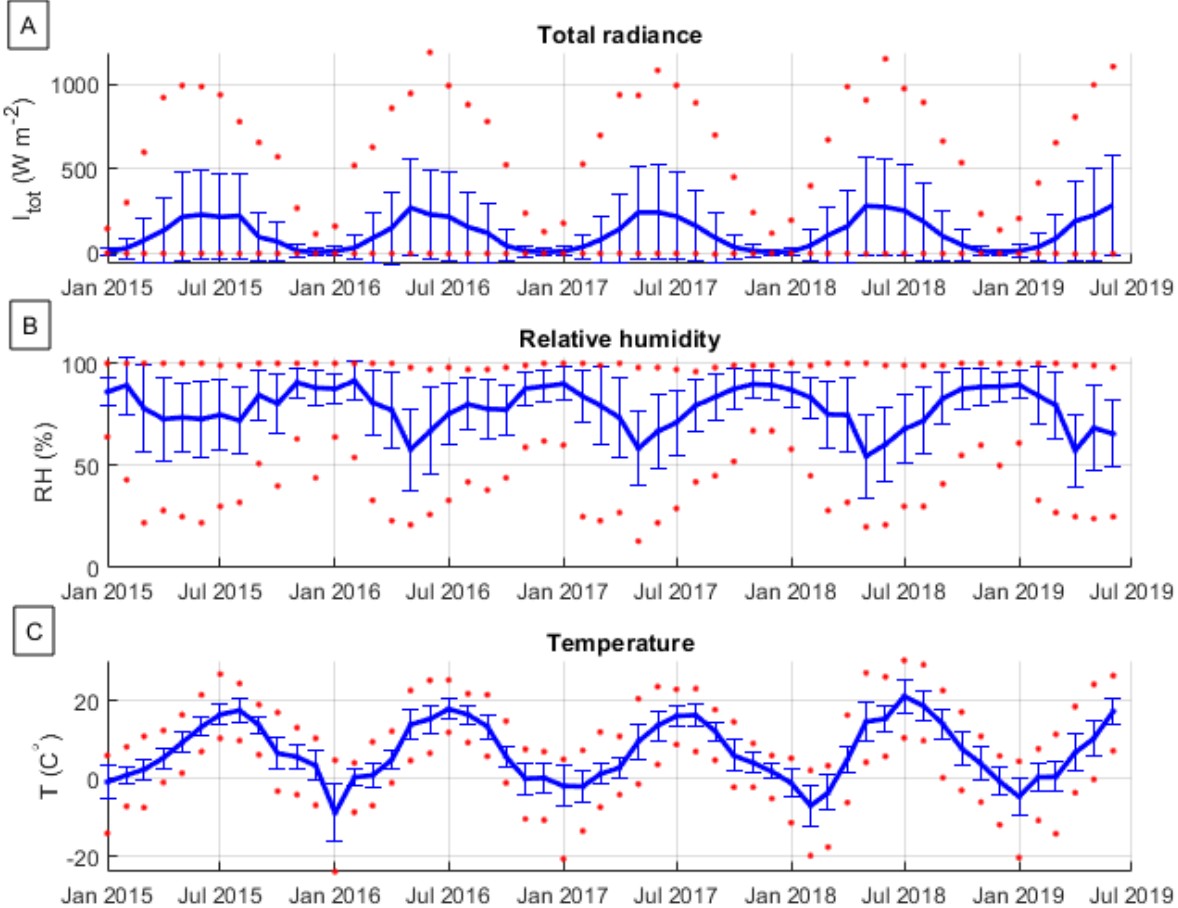

Figure 1: A presents monthly mean values for I$_{tot}$, B for RH, and C for T over the measurement period (2015-2019) in Helsinki. The constant blue line represents the monthly mean value. The bars show the standard deviation, and the red dots show the maximum and minimum values counted from hourly values.

### 2.4 Data processing

The data was processed in the following manner before being input into the PMF: Initially, outliers were identified and eliminated separately at both stations. Subsequently, the data was averaged on an hourly basis independently at each station. The data was then interpolated to 16-size bins at both locations. Finally, the data from the two sites was merged horizontally

into a single matrix with 32 bins in total. In more detail, strong outliers were removed from the DMPS data by calculating the total concentration and removing the data points that had a concentration ten times larger or smaller than the adjacent measurement points. These relatively relaxed outlier criteria were applied because the measurement site is less than a meter from the driving lanes, and therefore, variations in the concentrations can be expected due to passing cars. After removing the outliers, the size distribution data from the DMPS was averaged over 1-hour periods to minimize the effect of varying lengths of measurement cycles (approx. 8 min 40 s to 9 min 5 s at the SC and approx.. 9 min 50 s to 10 min 5 s at the UB). Additionally, this allowed for matching time stamps on the SC and UB data. Only 1-hour data points that had full coverage of data from both stations were included in the subsequent analyses. Averaging is not expected to affect the source profiles significantly, and average PNSD data has been used in PMF analysis in previous publications (e.g., Ogulei, Hopke, & Wallace, 2006).

The data analysis was done using a novel method of combing PNSD data from the two measurement stations horizontally so that PMF solved factors for both stations simultaneously. To the authors' knowledge, this approach has not been used before. In this approach, a single common factor is calculated for both stations, comprising 32 size bins. The initial 16 size bins are associated with the SC and the remaining 16 with the UB. Given that there is only one set of factors, the time series are identical for both stations, whereas the size distribution profiles vary between the sites. If a factor has a substantial local contribution at one of the sites but not at the other, then its profile would be pronounced at that station and near zero at the other. To be able to do this reliably and give even weight to data of both stations, the number of bins and the particle size limits needed to be scaled to the same. This was also necessary because growing the size of data files over a certain point would cause EPA PMF 5.0 program to crash during the analysis because of the program running out of memory. The data was reduced so that SC and UB PNSDs were presented in 16-size bins each. Data size bins have been modified for PMF in other studies. For example, Zhou et al. (2005), reduced 165 bins to 33 by averaging over five bins. In this study, bins were reduced so that a vector with an even lognormal bin width was created starting from 1 nm (lognormal width of 0.11 in this study). Then, the SC and UB data were interpolated linearly to this diameter vector so that the value for each new diameter point was given as a linear interpolation in a logarithmic x-axis between the nearest original diameters. The new size bin midpoint diameters were: 12.6, 16.2, 20.9, 26.9, 34.7, 44.7, 57.5, 74.1, 95.5, 123, 158, 204, 263, 339, 437, and 562 nm. The interpolation needs to be done on a logarithmic x-axis; otherwise, the interpolated concentration is overestimated with a negative derivative and underestimated with a positive derivative of the PNSD curve. This effect is demonstrated in supplement Fig. S1. This approach enabled us to give the same diameters to both sites regardless of the original size bins. This procedure has two drawbacks. One is that the data is levelled slightly, as the interpolated values are always between two original values. Therefore, the highest peaks are slightly lower and the lowest bottoms are slightly higher than in the original data. The second drawback is that some of the smaller changes in the PNSD may be lost. An example of reduced data compared to the original data is presented in supplement Fig. S2.

The decision to join the data together horizontally instead of doing the PMF analysis separately for the SC and UB was made because the idea in this study was to use only PNSD data as input data in PMF analysis and only to use the auxiliary data for identification of the factors. When the PMF analyses were done separately for the stations without additional data, the PMF

was able to split the measured PNSD into factors at each site. However, seemingly any number of factors could be fitted, with PMF only fractioning the measured PNSD to more sub-modes. Additionally, when PMF analysis was performed separately the attained factors had different modes and mode concentration in between the stations in all cases. This is not likely to resemble reality as the stations reside less than 1 km from each other and therefore somewhat similar background and long-range transport factors would be expected. Adding the data from the two sites together horizontally forces the PMF to find a common time series between the stations. This is beneficial in finding the common factors for the UB and SC as the time series of the common factors can be expected to be similar because of the small distance between the stations. On the other hand, joining the data horizontally does not force the same factor profiles for both sites. An additional problem of doing the PMF separately for the stations was that all factors at the SC site seemed to correlate strongly with the traffic diurnal cycle indicating that the traffic emissions are split between the different factors. The figures showing the 4, 5, and 6-factor solutions of the PMF analyses done separately for SC and UB are presented in supplemental material S3, S4, and S5 respectively. The negative side of merging the data horizontally can be expected to lower the total explained a fraction of the PNSD while the common time series are forced for both of the stations.

Source apportionment is typically conducted seasonally (winter, spring, summer, and autumn) among most long-term size distribution source apportionment analyses (Hopke et al., 2022). However, in this study, the data were analysed in one set to allow us to evaluate the changes in the contributions of various factors over the whole measurement period. Rose et al. (2021) stated that to maintain the representation of the total reliable concentration ($N_{tot}$), data coverage needs to exceed 50% on the seasonal level and 60% on the annual level; for the reliable evaluation of diurnal variation, yearly coverage of 75 % was required. The seasonal coverages of the overlapping data for both sites are presented in Table 2. Notably, the coverages for the first and last seasons (i.e., winter 2015 and summer 2019) were low, as the measurement period started and ended in the middle of the seasons.

Table 2: Seasonal overlapping data coverage for UB and SC sites.

|  | Winter (Dec-Feb) | Spring (Mar-May) | Summer (Jun-Aug) | Autumn (Sep-Nov) |
| --- | --- | --- | --- | --- |
| **2015** | 14% | 81% | 92% | 92% |
| **2016** | 62% | 70% | 88% | 90% |
| **2017** | 86% | 49% | 89% | 91% |
| **2018** | 73% | 48% | 84% | 93% |
| **2019** | 85% | 88% | 5% | - |

## 2.5 PMF

Developed by Paatero (1997), PMF is a multi-derivative method that is widely used in environmental sciences to apportion the sources of the measured data. PMF is a least-squares method based on the fact that the matrix X with dimensions n x m can be presented as a product of two matrixes: A, with dimensions n x y, and B, with dimensions of y x m. This can be used so that the matrix n x m is the measured result matrix with n observations and with m species, or, in the case of PNSD, particle size bins and y can be set as the number of independent sources.

In estimating error, the methodology was established by Ogulei, Hopke, Zhou, et al. (2006) and further developed by Rivas et al. (2020). The measurement uncertainties ($\sigma_{ij}$) were calculated with the following equation:

$$\sigma_{ij} = \alpha(N_{ij} + \underline{N}_j)$$

where $\alpha$ is the arbitrary constant, similar to Rivas et al.'s (2020) work (0.02 for the SC and 0.022 for the UB), and $N_{ij}$ is the concentration of sample i in size bin column j, and the $\underline{N}_j$ is the arithmetic mean of concentration in size bin j. The overall

uncertainty was calculated using the following equation:

$$S_{ij} = \sigma_{ij} + C_3 \cdot N_{ij}$$

where $\sigma_{ij}$ is measurement uncertainty, $C_3$ is an arbitrary constant that was set in this study to 0.1 for both the UB and SC, and $N_{ij}$ is the concentration of bin j of sample i.

Wiedensohler et al. (2012) stated that concentration measurement errors seem to be approximately double for particles in the

size range of 200-800 nm compared to 20-200 nm. Therefore, in this study, the measurement errors are corrected for these particle sizes by doubling the $\alpha$ factor.

The most reasonable solution to fit the data was a five-factor solution based on the testing to produce results with the most meaningful physical interpretation and reasonable residuals. The robustness of the solution was tested using five different random seeds as starting points. Performing analysis with a larger seed number sometimes caused the program to crash. In

addition to using random sees a displacement analysis was performed on the solutions. The results of the displacement analysis showed no drop in Q values or swaps in any of the analyses. The factors were identified as fresh traffic (TRA1), slightly aged traffic (TRA2), secondary combustion aerosol (SCA), secondary aerosol (SecA), and long-range transported aerosol (LRT). The dispersion-corrected PMF results were also calculated for the five factors for comparison, and the difference in the results calculated without the dispersion correction was found to be negligible (Dai et al., 2021). The differences between the normal

and dispersion normalised PMF were evaluated based on the Pearson correlation coefficients between workday diurnals (>0.98 for all factors), weekend diurnals (>0.98 for all factors), monthly contributions (>0.96 for all factors) and factor profiles (>0.97 for all factors). In dispersion correction, the original measurement data is normalized by the ventilation coefficient which is the height of the boundary layer times the average wind speed during the period. The goal of the dispersion correction is to reduce the inaccuracy in the source apportion caused by the dispersion of aerosol in the atmosphere (Dai et al., 2021). The

results calculated with dispersion normalization are presented in supplement Fig. S6. Fig. 2 shows the mean residuals, mean scaled residuals, mean relative residuals, and Q/Q$_{exp}$ values for the different number of factors between 2 and 10. At the chosen

five-factor solution, the mean relative residual was only around 2.8% on average. The residual and $Q/Q_{exp}$ values decrease continuously as the number of factors increases. However, for scaled residuals, mean relative residuals and $Q/Q_{exp}$ values, the decrease is smaller after increasing the number of factors past five. This is an indication that five is an acceptable number of

factors. Additionally, the neighbouring solutions of four and six are presented in supplemental Figures S7 and S8, respectively. The four-factor solution merges the factors described later in this paper (SCA and SecA). In a five-factor solution, these two have notably different diurnal profiles, and therefore, merging them is not sensible. The six-factor solution presented in S8 splits the SCA into two factors that have very similar diurnal profiles and contributions throughout the year, and therefore, they are likely to be from the same source. In supplemental material S9, the average relative residuals with the standard

deviation are presented for each size bin. Also, the figures showing the regressions between the modelled and measured concentrations after interpolation are presented in supplemental material S10. The average difference between these was 12.7 % at SC and 6.7 % at UB and the temporal patterns for the difference have been presented in supplemental material S11.

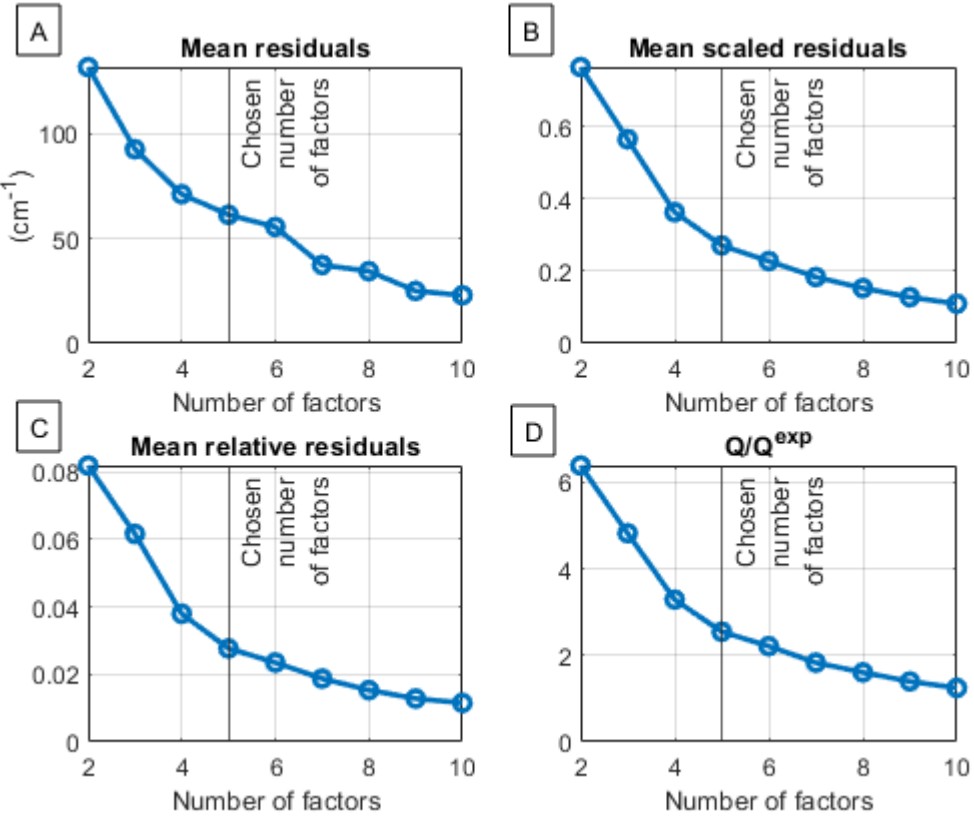

**Figure 2: Mean residuals (A), mean scaled residuals (B), mean relative residuals (C), and Q/Q$_{exp}$ values (D) for the different number**
**of factors between 2 and 10. The mean residuals presented have been calculated size-wise as an average over the unified dataset from the SC and UB measurement locations.**

**2.6 Trend analysis**

Time series of PMF factors were fitted with trends using the Theil-Sen estimator established by Theil (1950) and further developed by Sen (1968). The base-level concentrations at the beginning of the measurement period were calculated using the slope determined by the Theil-Sen estimator with each data point and, counting the median. Because factors had clear seasonal, variance the Theil-Sen estimator was plotted in two ways. One involved using the seasonal Theil-Sen estimator, in which only data from the same months is compared in forming the estimate. The other way was first removing seasonality from data using the seasonal-trend decomposition procedure presented by Cleveland et al. (1990). The seasonal trend removal was needed because the reliability of the results was evaluated using the Mann-Kendall test for monotonic trends (Mann, 1945), which can not be done with seasonal data. The results are presented in Table 5. Figures showing the trend decomposition for the factors and the fitted Theil-Sen estimators are presented in the supplemental material. The trend decompositions for TRA1, TRA2, SCA, SecA, and LRT are presented in S12, S13, S14, S15, and S16, respectively. Additionally, the fitted Theil-Sen estimators are presented for TRA1, TRA2, SCA, SecA, and LRT in S17, S18, S19, S20, and S21 respectively. Notably, the trends calculated using the seasonal Theil-Sen estimator and the Theil-Sen estimator calculated from data without seasonal variability were almost identical, increasing the confidence in using seasonal trend decomposition for the data (Table 5).

**3 Results and discussions**

**3.1 General description of PNSD**

Particle PNSD was found to be noticeably different between stations. The time series of the daily average PNSD in each year are presented in Fig. 3. Figure 3k presents the Pearson correlation coefficient as a function of particle size when the daily PNC in each of the 16-size bins are compared between the SC and UB. The observed PNSD at the SC in the size range of 12.6 to 562 nm contained significantly more nanosized (< 100 nm) particles on many occasions, as well as higher overall particle concentrations compared to the UB. Notably, the correlation was higher for the larger particle sizes, indicating that if the smallest particles are disregarded, the time series would be relatively similar. The higher overall PNC at the SC, higher correlation above 200 nm particles, and lower correlation below 200 nm particles compared to the UB indicate that there was at least one local source producing a lot of nanosized particles at SC that was missing at UB.

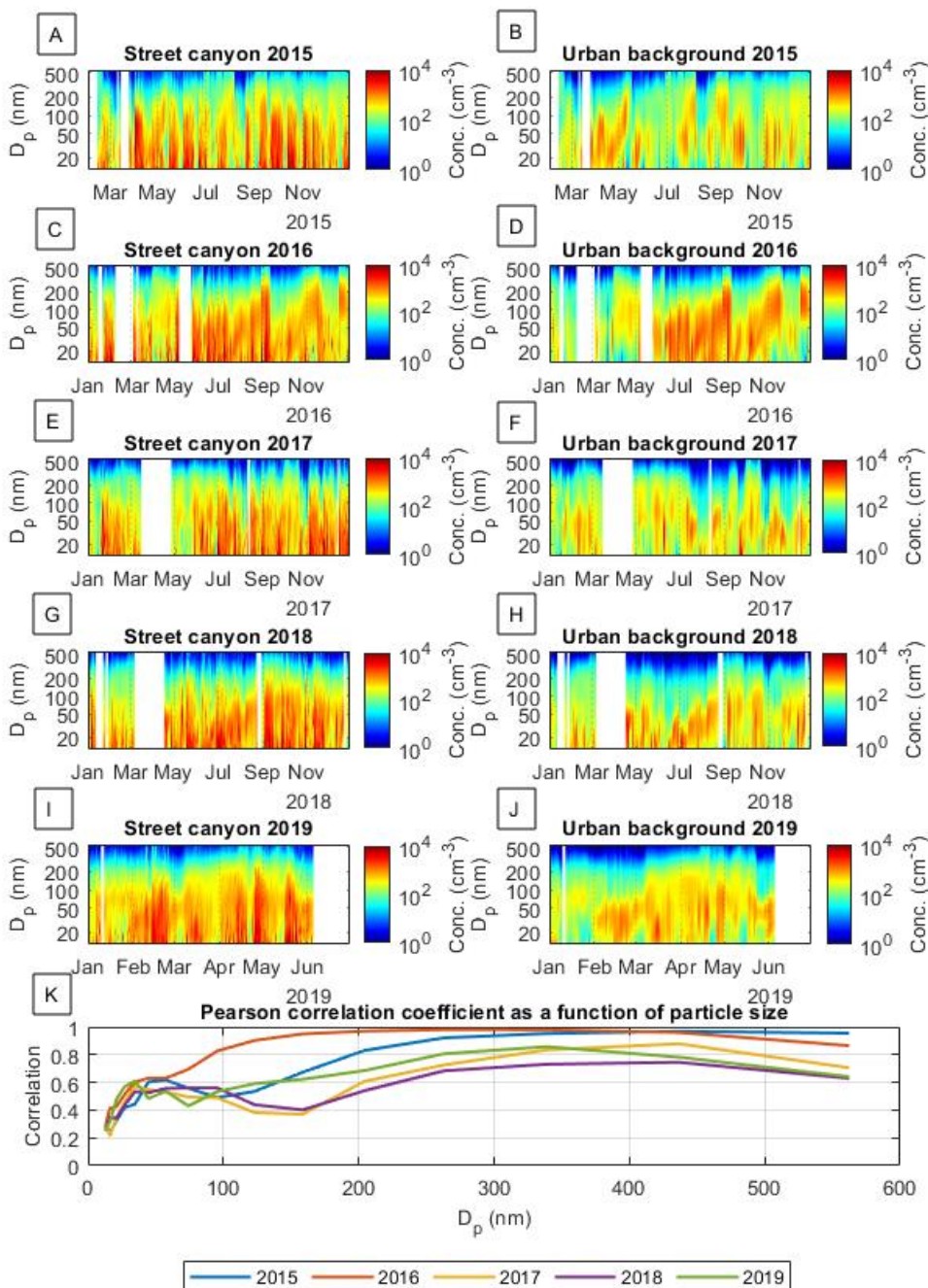

**Figure 3: Time series of daily average PNSD for SC and UB for each year: 2015 (A and B), 2016 (C and D), 2017 (E and F), 2018 (G and H), and 2019 (I and J). The data used is reduced to 16-size bins. The particle diameter ($D_p$) is presented on the y-axis, the x-axis presents the time, and PNC ($cm^3$) is shown by logarithmic color scale. The yearly correlation between the UB and SC stations (Pearson correlation coefficient) is presented in the bottom plot for the various particle sizes and daily mean concentrations.**

### 3.2 Identification of factors

The most reasonable solution in the PMF analysis was a five-factor solution. The factors were identified based on the diurnal profiles, annual variation of the factors, and comparison to the available auxiliary data measured at the UB and SC sites (trace gases, particle chemistry, PM mass concentrations, and meteorology). The factor profiles and identification variables, their
diurnal contributions for workdays and weekends, and monthly contributions are presented in Fig. 4 and Table 3. The contributions depicted in Fig. 4 represent the scaling factors applied to each source profile at specific times. For instance, if the contribution at a certain time is two, the corresponding source profile is scaled by a factor of two at that moment. The source profiles and their contributions are normalized such that the mean contribution from each source averages to one over the measurement period. These contributions were also calculated for each factor separately in supplemental material S22 for
different wind directions.

Table 3: PMF factors, their size modes, correlating variables, and identification.

| Factor | PN size mode (nm) | Important correlating variables | Identification arguments |
|---|---|---|---|
| TRA1 | 12.6 | BC, NO, NO$_2$, and NO$_x$ at SC | Particle size and diurnal profile similar to traffic intensity. Correlation with traffic tracers. |
| TRA2 | 16.2 | NO$_x$ and NO at UB | Diurnal profile is similar to traffic. Slightly behind TRA1 |
| SCA | 44.7 | NO$_x$ at UB, m/z 60 | Delayed peak after TRA1 and TRA2 correlation with NO$_x$ at UB |
| SecA | 74.1 | Total organics, m/z 43 | No difference between workdays and weekends, highest concentrations in summer. |
| LRT | 204 | PM$_{2.5}$, SO$_4$, NO$_3$ and total organics | Correlations with variables related to LRT and minimal diurnal profile |

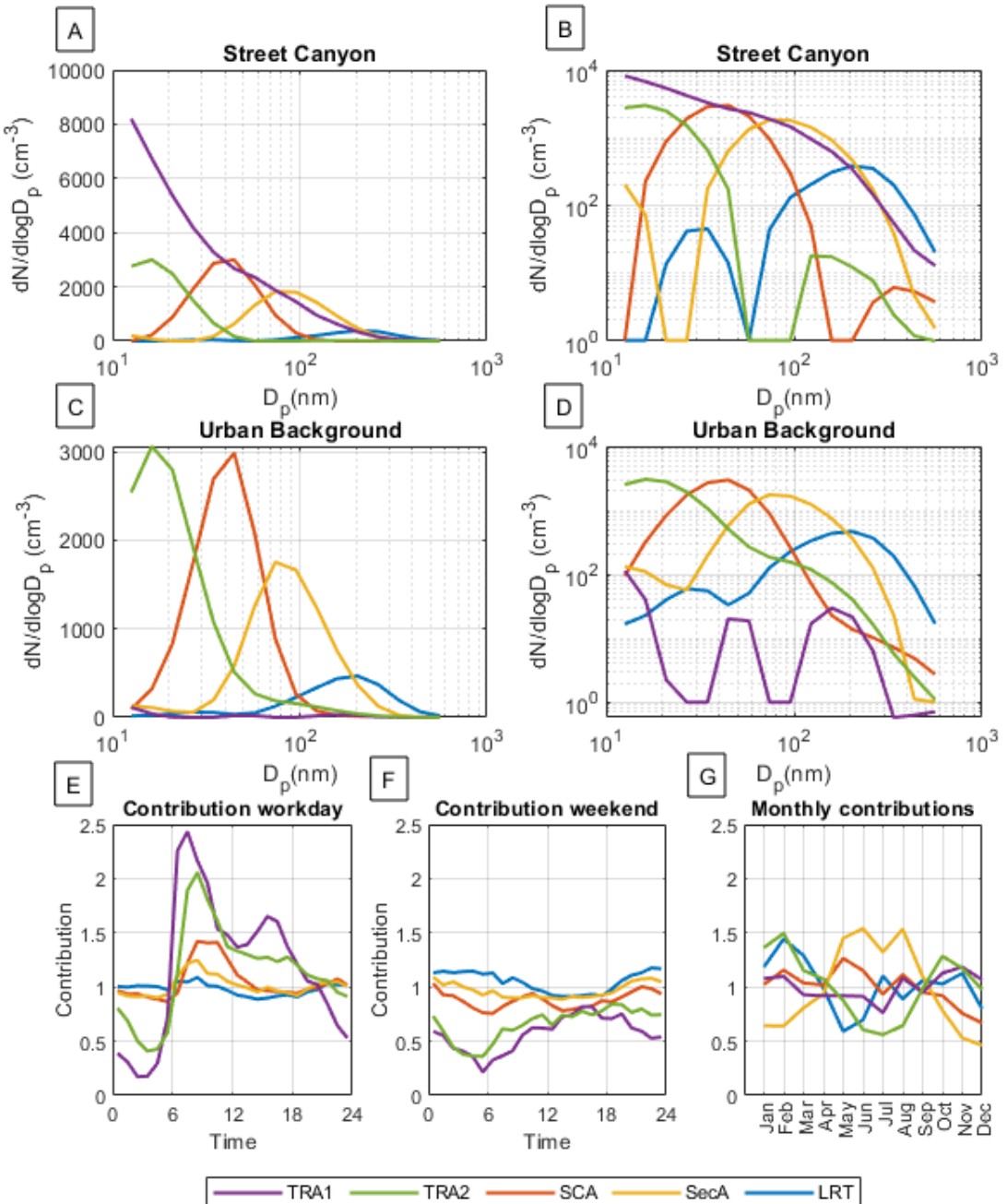

**Figure 4: PMF factors presented for both stations on linear (A for SC, C for UB) and logarithmic x-axis. E presents the hourly relative contributions during workdays, F during weekends, and G the average monthly contributions. Note that the linear scales for plots A and B are different. The value presented in contribution figures is the factor with which to multiply the factor profile at any current time to get the total contribution. The average for the contribution factor is 1 over the whole measurement period for all the factors.**

TRA1 was interpreted to represent particles that originated from local traffic emissions in the immediate proximity of the measurement station. This factor was the dominant factor in the SC, while it was almost zero at the UB, which was located on a hill over 100 m from the nearest busy road. TRA1 had the highest number of particles in the smallest measured particle size (12.6 nm) and the second mode at around 50-60 nm. Possibly, a third mode at 100-200 nm can be seen as a tail in the log-log plot (Fig. 4b). Similar non-volatile modes at 10 nm and 70 nm particle diameters have been reported for laboratory measurements for modern gasoline cars (Karjalainen et al., 2014). During weekdays, TRA1 had a distinctive diurnal profile, similar to BC and $NO_x$; these are often related to traffic emissions, with the largest peak during the morning rush hour and the second, slightly lower peak during the afternoon rush hour (Fig. 5a). TRA1 had significantly lower contributions during weekends, but a high correlation with NOx and BC (Fig. 5b). Overall, the linear relationship between the variables (Pearson correlation coefficient R) for TRA1 with BC (AE33 with 880 nm) and $NO_x$ were 0.76 and 0.85 at the SC, respectively. TRA1 had also a high correlation with $NO_2$ and NO at SC (Table 4).

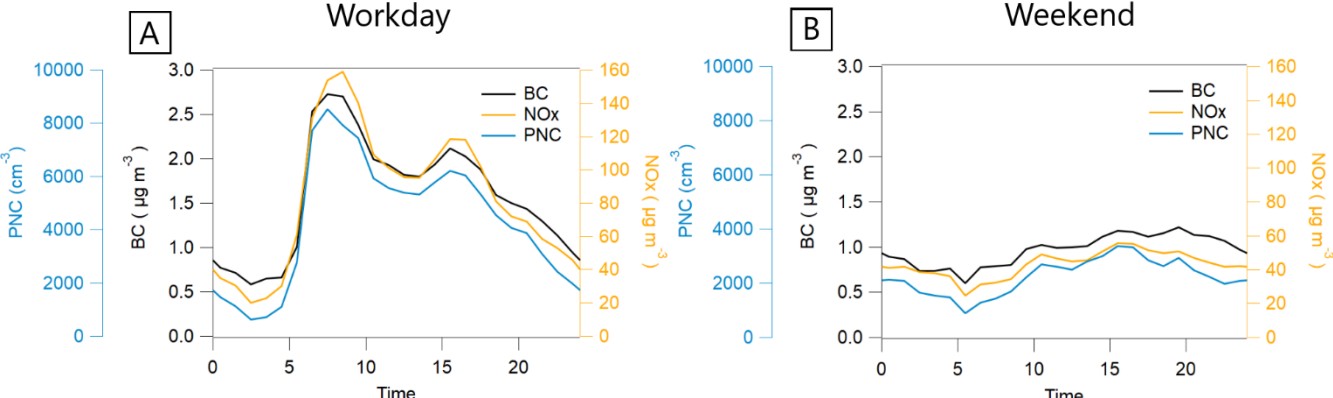

**Figure 5: Diurnal profiles for TRA1 factor-related PNC concentrations at the SC, along with $NO_x$ and $BC_{880}$ concentrations presented separately for workdays (A) and weekends (B).**

TRA2 was interpreted as a slightly aged traffic-related factor. Atmospheric aging of aerosols is expected to increase the mode particle size of PNSD due to the oxidation of gaseous volatile organic compounds (VOCs) into compounds with lower volatilities. These oxidized compounds then condense on existing particles, making them larger. Furthermore, smaller particles experience greater diffusion losses. Consequently, we can expect a shift in the mode particle size toward larger particles during aging. The aged nature of the factor was concluded as shown in Fig. 4e; the morning rush hour peak of TRA2 is observed 1 hour later compared to TRA1. TRA2 also has similar contributions at the SC and UB, and therefore, the TRA2 was considered to be slightly aged, as the road is further away (100 m) from the UB station. Additionally, the mode particle size was larger for TRA2 compared to TRA1, with a maximum mode particle size of 16.2 nm at both stations. TRA2 also displayed a diurnal trend that matches the traffic pattern, having a peak during the morning rush hour and elevated concentrations for the rest of the working hours of the day (Fig. 4e). The morning peak was noticed approximately 1 hour later than the TRA1 factor. Therefore, TRA2 can be considered slightly aged, regionally processed traffic emissions. The most significant correlations of TRA2 with auxiliary data were with $NO_x$ and NO measured at the UB (Table 4). Additionally, TRA2 correlated moderately

with $NO_x$ and NO measured at the SC. This moderate correlation was expected, as the $NO_x$ and NO measured at the UB are the background levels that are also measured at the SC. However, in addition, the $NO_x$ and NO concentrations at the SC are strongly influenced by the immediate traffic emissions, and therefore, the correlation of TRA2 with the concentrations at the SC is lower than with the concentrations at the UB. This also supports the slightly aged character of TRA2, as the SC site was dominated by immediate traffic-caused emissions (Fig. 4a). The TRA2 was more concentrated during colder months, which might be because in cold T VOCs condense more efficiently on existing particles (Fig. 4g). In addition, the boundary layer is shallower during cold months enhancing the accumulation of primary pollutants.

The SCA factor had a peak particle size of 44.7 nm at both sites and was interpreted as a secondary aerosol originating from combustion processes (i.e., of liquid fuel such as diesel, oil solid fuel such as biomass and coal, or gas). SCA had relatively weak correlations with the primary traffic emissions (e.g., $NO_x$, BC, CO, m/z 57) data, as could be expected for atmospherically processed aerosol. Of these m/z 57 ($C_4H_{9+}$, $C_3H_5O+$) is a part of the HOA mass fraction that is linked to traffic exhaust emissions (Crilley et al., 2013; Crippa et al., 2012; Daellenbach et al., 2016; Mohr et al., 2012). The strongest Pearson correlation coefficient of 0.56 was observed between SCA and $NO_x$ at the UB site (Table 4). SCA and $NO_x$ at the UB site also had similar diurnal patterns on working days and weekends (Fig. 6). The highest SCA peak was seen approximately 3 hours later than for the TRA1 factor, indicating that the SCA factor included traffic emissions that had been aged/processed a couple of hours in the atmosphere. SCA was found to have an evening peak in addition to the morning rush hour peak (Fig. 4e). The evening peak was more pronounced during weekends, which indicates possible contributions from biomass combustion (Fig. 4f). In an earlier study, BC originating from biomass combustion was shown to contribute $15 \pm 14\%$ at the SC and between $41 \pm 14$ and $46 \pm 15\%$ of the BC in residential/detached house areas (Helin et al., 2018). To support this, the diurnal trends of SCA and organic fragments at m/z 60 (Q-ACSM) at the SC were plotted (Fig. 6). The fragments at m/z 60, particularly its fraction of the total OA, have widely been used as a marker for primary wood combustion emissions (Alfarra et al., 2007). The shape of the m/z 60 diurnal profile was similar to the SCA diurnal profile during the workdays and weekends strengthening the assumption of wood combustion contribution to SCA. An important thing to note is that the overall correlation with the m/z 60 was still relatively low (Table 4). The similar rush hour peak of m/z 60 to that of SCA was slightly surprising as the m/z 60 is usually related to biomass combustion and not traffic. The annual variation of SCA is small (Fig. 4g). Likely because although during the wintertime, the amount of biomass combustion increases the amount of sunlight is low, limiting SOA formation, whereas during summer, the amount of biomass burning is lower, but the amount of sunlight increases, thus enhancing SOA formation. In contrast, traffic emissions remain stable throughout the whole year.

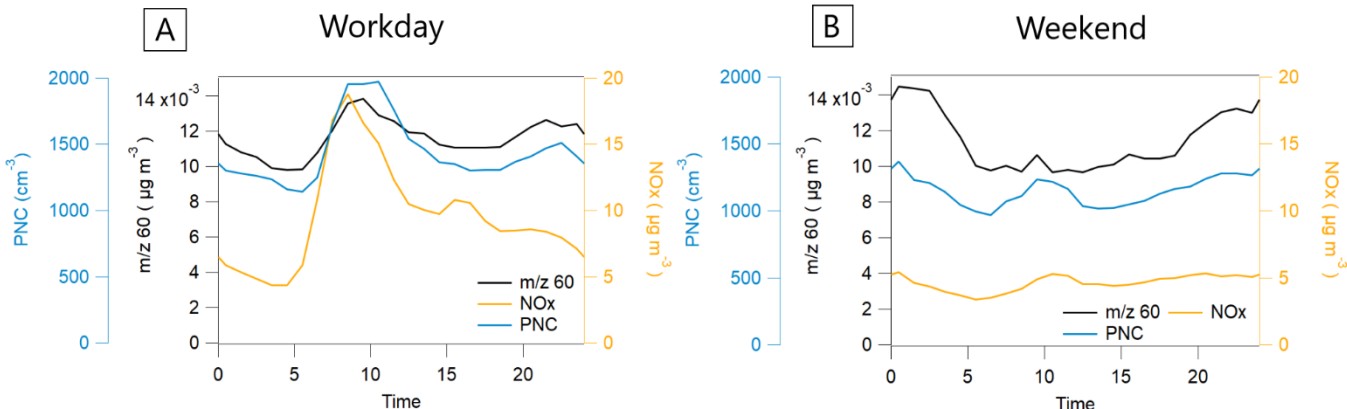

**Figure 6: Diurnal profiles for SCA factor-related PNC concentrations at the SC, along with NOₓ concentration at the UB station and organic fragments at m/z 60 concentrations at the SC during workdays (A) and weekends (B).**

The SecA factor had a peak particle size of 74.1 nm at both sites and was interpreted as an aged, photochemically formed secondary aerosol from biogenic and anthropogenic precursors. This assumption is based on the negligible difference in diurnal profiles between workdays and weekends and elevated contribution during the summer months with the highest total radiance (Fig. 1a and 4g). Additionally, the strongest correlations of the SecA factor were with total organics and m/z 43 (Table 4). The m/z 43 has been associated with less oxidized secondary organic aerosol (Chen et al., 2022) Anthropogenic and biogenic VOCs

are shown to be important SecA precursors in a traffic environment (Saarikoski et al., 2023). In addition, the SecA had moderate correlations with m/z 57 and m/z 60 (Table 4). The reason for this is likely to be the high total amount of organics as the m/z 57 and m/z 60 do not refer to the relative fraction of the total organics but absolute concentrations of the mass fraction and therefore they might be elevated with the higher total organic mass in particles. Surprisingly, the SecA factor also somewhat correlates with BC (Table 4), possibly indicating that BC particles that are ubiquitous in traffic environments might

act as cores for or be mixed into these SecA particles.

The long-range transport (LRT) factor had a peak particle size of 204 nm at both sites and is interpreted as a long-range transport because of its strong correlations with $PM_{2.5}$, $SO_4$, $NO_3$, and organics (Table 4). The correlation was even higher (0.80) with the sum of $NO_3$ and $SO_4$ at the SC. Typically higher concentrations of accumulation mode particles have been observed during LRT events (Timonen et al., 2008). Furthermore, Niemi et al. (2009) showed that relatively high

concentrations of inorganic ions, especially $SO_4$, $NH_4$, and BC, are typically observed during LRT events. The correlation of LRT with $NH_4$ was relatively high, but the correlation with BC was quite low (Table 4). The reason for the low correlation with BC might be due to local sources of BC (e.g., traffic and the short atmospheric lifetime of BC; (Cape et al.,)). Additionally, Niemi et al. (2009) did not report high concentrations of $NO_3$ during the LRT episodes, likely because of evaporation losses of ammonium nitrate from the Teflon filters. However, more recent studies with online analysis of $NO_3$ have linked elevated

$NO_3$ concentrations to LRT episodes in the area (Harni et al., 2023; Barreira et al., 2021; Pirjola et al., 2017). Also, elevated $PM_1$ and $PM_{2.5}$ concentrations have been related to the LRT episodes in the area (Harni et al., 2023; Barreira et al., 2021; Niemi et al., 2009; Pirjola et al., 2017). The LRT factor had a moderate correlation with m/z 60. This is likely to be caused by

the large PM of LRT particles and therefore higher total organic mass. Additionally, the correlation with m/z 60 might indicate the contribution of remote biomass burning to the LRT factor.

**Table 4: Pearson correlation coefficients of the PMF solution factors (LRT, SCA, SecA TRA1, and TRA2) with other factors and other measured parameters ( with NO$_x$, O$_3$, SO$_2$, NO, and CO from the UB station; NO, NO$_x$, NO$_2$, PM$_{10}$, PM$_{2.5}$ PM$_{coarse}$, O$_3$, CO, BC$_{880}$(AE33), CO$_2$, PM$_{tot}$, m/z 43, m/z 57, m/z 60, Chl, NH$_4$, NO$_3$, ORG$_{tot}$ and SO$_4$ from SC, I$_{tot}$, and RH from the Kumpula weather station, and T from the Kaisaniemi weather station.**

| | LRT | SCA | SecA | TRA1 | TRA2 |
|---|---|---|---|---|---|
| NO$_x$ (UB) | 0.19 | 0.56 | 0.46 | 0.33 | 0.48 |
| O$_3$ (UB) | -0.19 | -0.25 | -0.08 | -0.24 | -0.32 |
| SO$_2$ (UB) | 0.05 | 0.21 | 0.09 | 0.04 | 0.13 |
| NO (UB) | 0.12 | 0.44 | 0.34 | 0.2 | 0.42 |
| CO (UB) | 0.33 | 0.32 | 0.23 | 0.17 | 0.34 |
| NO (SC) | 0.05 | 0.36 | 0.27 | 0.82 | 0.32 |
| NO$_x$ (SC) | 0.06 | 0.38 | 0.31 | 0.85 | 0.33 |
| NO$_2$ (SC) | 0.08 | 0.39 | 0.36 | 0.8 | 0.3 |
| PM$_{10}$ (SC) | 0.12 | 0.23 | 0.23 | 0.42 | 0.2 |
| PM$_{2.5}$ (SC) | 0.74 | 0.24 | 0.45 | 0.36 | 0.11 |
| PM coarse (SC) | -0.07 | 0.11 | 0.06 | 0.22 | 0.13 |
| O$_3$ (SC) | -0.09 | -0.2 | -0.07 | -0.39 | -0.28 |
| CO (SC) | -0.06 | -0.02 | -0.11 | -0 | 0.14 |
| BC$_{880}$ (SC) | 0.26 | 0.35 | 0.44 | 0.76 | 0.24 |
| CO$_2$ (SC) | 0.03 | 0.02 | 0.1 | 0.11 | -0.04 |
| PM$_{tot}$ (SC) | 0.09 | 0.26 | 0.31 | 0.46 | 0.16 |
| m/z 43 (SC) | 0.54 | 0.18 | 0.63 | 0.23 | -0.03 |
| m/z 57 (SC) | 0.35 | 0.36 | 0.52 | 0.59 | 0.16 |
| m/z 60 (SC) | 0.64 | 0.24 | 0.5 | 0.21 | 0.07 |
| Chl (SC) | 0.25 | 0.04 | 0.07 | 0.02 | 0.03 |
| NH$_4$ (SC) | 0.62 | 0.04 | 0.12 | 0.07 | -0.01 |
| NO$_3$ (SC) | 0.69 | 0.16 | 0.22 | 0.15 | 0.03 |
| ORG$_{tot}$ (SC) | 0.64 | 0.2 | 0.65 | 0.25 | -0.03 |
| SO$_4$ (SC) | 0.71 | 0.01 | 0.11 | -0.03 | -0.05 |
| I$_t$ot | -0.12 | 0.05 | 0.16 | 0.05 | -0.02 |
| RH | 0.24 | -0.05 | -0.09 | 0.07 | -0.06 |
| T | -0.03 | -0.06 | 0.32 | -0.03 | -0.3 |
| LRT | 1 | 0.01 | 0.26 | -0 | -0.04 |
| SEC | 0.01 | 1 | 0.37 | 0.21 | 0.3 |
| SOA | 0.26 | 0.37 | 1 | 0.13 | 0.02 |
| TRA1 | -0 | 0.21 | 0.13 | 1 | 0.23 |
| TRA2 | -0.04 | 0.3 | 0.02 | 0.23 | 1 |

## 3.3 Monthly average contributions and trends of factors

Figure 7 represents the time series for the contributions of the PMF factors to PNC at the SC and UB sites. The average monthly contributions at the SC site were 52%, 15%, 17%, 13%, and 3% for TRA1, TRA2, SCA, SecA, and LRT, respectively. For the UB, the corresponding monthly average contributions were 1%, 36%, 34%, 23%, and 7% for TRA1, TRA2, SCA, SecA, and LRT, respectively. TRA1 was seen to be the main contributor to PNC at the SC, while at the UB, the PNC was usually dominated by the slightly aged combustion-related factors, TRA2 and SCA, with quite similar contributions, as could be expected for stations situated next to the road and 100 m away from the road. During summertime, SecA also made a contribution that was roughly even with those of TRA2 and SCA at both stations, highlighting the importance of secondary aerosol formation even in urban environments. Barreira et al. (2021) described the increased contribution of organics aerosol mass during summertime in Helsinki. Similar contributions of traffic-related aerosols either fresh (46%) like TRA1 in this study or aged (27 %) like TRA2+SCA (sum 28 %) have been reported in the roadside environment  (Al-Dabbous & Kumar, 2015).

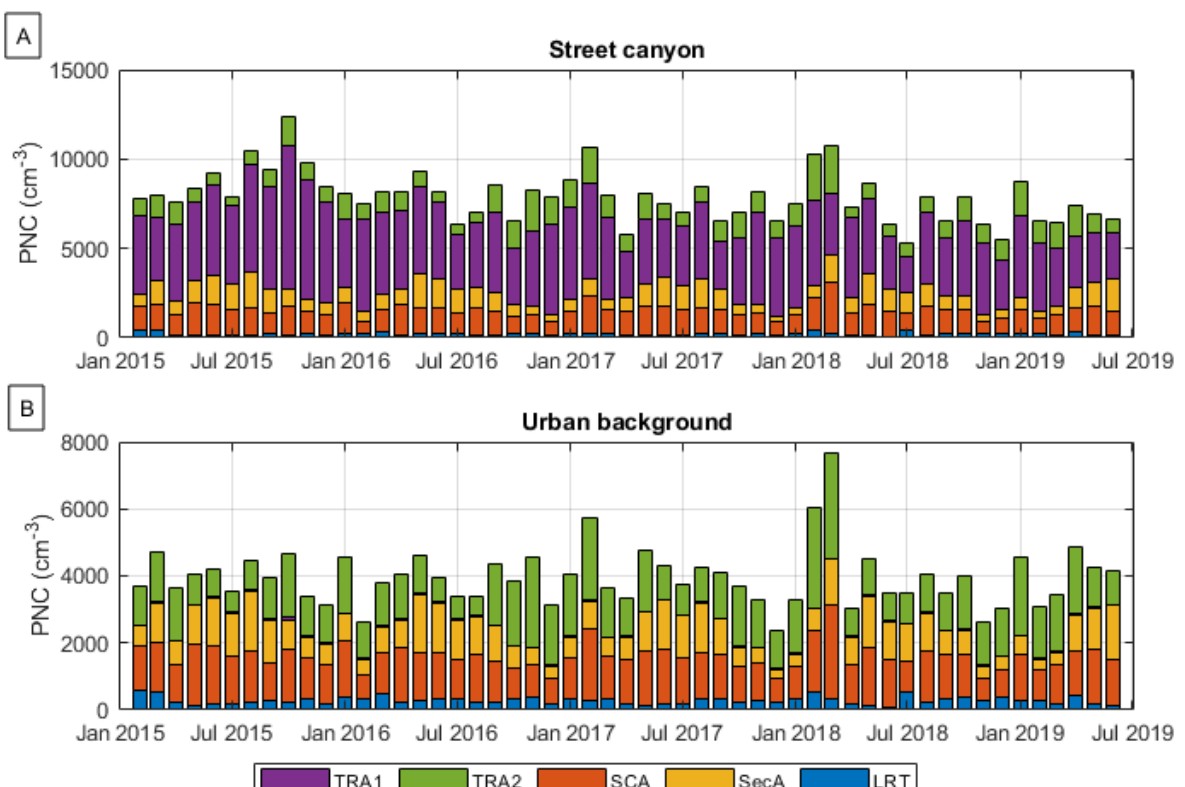

Figure 7: Contribution of various factors to PNC at the SC (A) and UB (B) sites.

The contributions of factors at the SC and UB stations were also calculated in terms of particle volume. Notably, the contributions of the various factors to volume concentrations were different when compared to contributions to the PNC (Fig. 8). The average monthly contributions at the SC were 28%, 1%, 4%, 26%, and 41% TRA1, TRA2, SCA, SecA, and LRT,

respectively. For the UB, the monthly average contributions were 1%, 5%, 7%, 29%, and 59% for TRA1, TRA2, SCA, SecA, and LRT, respectively. Compared to PNC, the contributions of TRA1, TRA2, and SCA decreased, whereas those of SecA and especially LRT increased. The largest contributor to volume concentration was LRT, followed by SecA at the UB. At the SC,

the second largest contributor to particle volume during summer months was also SecA but during winter the second largest contributor was TRA1. The contributions of LRT to volume concentration varied greatly from month to month at both stations. The months of the highest concentrations varied between years, highlighting the event nature of this factor, as singular strong events can increase LRT contributions. This is in contrast to for example TRA1, which shows little month-to-month variation, and the concentrations stay relatively stable between the different months in Fig. 7 and Fig. 8.

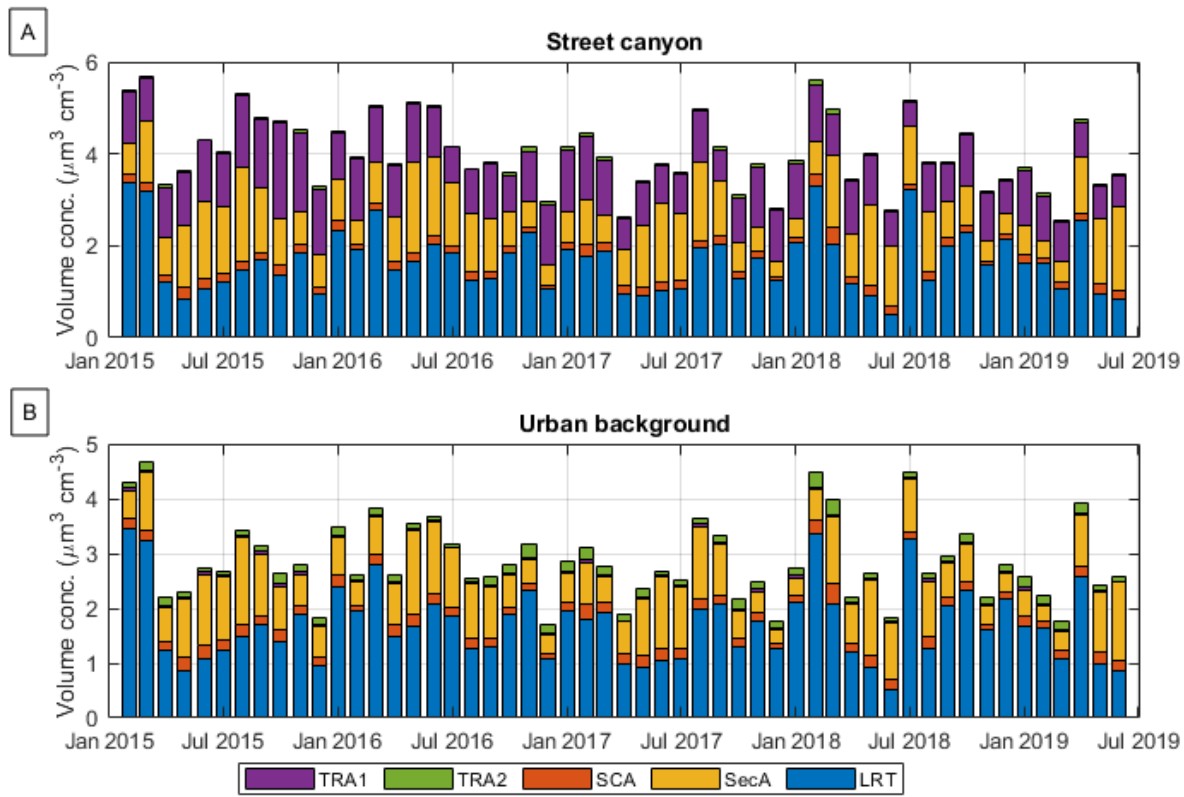

**Figure 8: Monthly average volume-based contributions of various factors at the SC (A) and UB (B) stations during the measurement period.**

Three statistically significant trends in the factor concentrations were found: decreasing trends of -9.5% and -6.5% yearly for TRA1 and SecA, respectively, and an increasing trend of 6.4% yearly for TRA2 (Table 5). The decreasing trend of TRA1

seems to imply that the primary emissions from traffic have been decreasing over the years, supported by the fact that the vehicle fleet renewed rapidly in Finland between 2015 and 2019. For instance, the proportions of low-emission (EURO 6/VI grade) vehicles in driven kilometers have increased in different vehicle classes in Finland as follows: cars from 6% to 29%, vans from 1% to 25%, and trucks from 7% to 31% (VTT's LIPASTO calculation system for traffic exhaust emission in

Finland). The proportion of low-emission city buses has increased particularly quickly in the Helsinki metropolitan area, from
420 17% to 59% over the course of 5 years (statistics from the Helsinki Regional Transport Authority HSL). The impact of bus emissions is significant at the Mäkelänkatu SC site, since there is a bus lane very close (0.5 m) to the air quality monitoring station. Notably, TRA2 has increased, possibly due to the change in the engine and after-treatment techniques of the vehicles, emphasizing the significance of atmospheric processes interacting traffic emissions.

The decrease of the SecA factor is more complex, to explain as it was speculated to have both anthropogenic and biogenic sources. The latter is closely connected to relatively stable biogenic sources and meteorology (T and $I_{tot}$) and the former to the fast development of cleaner engine and after-treatment technologies driven by new emission limits. In the previous chapter, SecA concentrations were shown to indicate possible correlation with BC. One possible reason for this was suggested to be that the BC particles could act as cores for SecA. In fact, the decreasing trend was similar to earlier presented BC trends
(decrease between -10& and -6% $yr^{-1}$) in different environments (traffic, urban background, and regional background) in Finland (Luoma et al., 2021). No statistically significant trend was found for SCA or LRT, although a slight decrease was indicated for both. There could be many reasons for this. Although traffic emissions have decreased (e.g. Barreira et al., 2021), biomass combustion for residential heating has increased lately. LRT emissions are mainly affected by meteorological conditions and can vary a lot between years. We note that more data (years) is needed to see if the trends found in this study
were real as the larger the sample size is, the better the hypothetical test is. Table 5 presents all the results of the trend analysis.

**Table 5: Seasonal Theil-Sen estimators and Theil-Sen estimators calculated from data with seasonality removed with confidence levels and base-level concentrations at the beginning of the measurement period for TRA1, TRA2, SCA, SecA, and LRT. The trend in the table is significant if the significance level is above 95% (p-value < 0.05).**

| | | TRA1 | TRA2 | SCA | SecA | LRT |
|---|---|---|---|---|---|---|
| **Seasonal Thel-Sen estimator** | Base concentration ($cm^{-3}$) | 5175 | 953 | 1445 | 1073 | 214 |
| | Yearly change ($cm^{-3}$) | -493 | 61 | -32 | -70 | -6 |
| | Relative change (%) | -9.5 | 6.4 | -2.2 | -6.5 | -2.7 |
| **Theil-Sen estimator for data with no seasonality** | Base concentration ($cm^{-3}$) | 5149 | 953 | 1407 | 1074 | 210 |
| | Yearly change ($cm^{-3}$) | -471 | 62 | -11 | -70 | -5 |

| Relative change (%) | -9.1 | 6.5 | -1 | -6.6 | -2.3 |
|---|---|---|---|---|---|
| Significance | yes | yes | no | yes | no |

## Conclusions and summary

Particle size is one of the most important parameters of atmospheric particles in terms of health and climate effects. In this study, the origin and characteristics of particle PNSD were investigated in Helsinki, southern Finland. The measurements were carried out at two sites, an UB and a SC site, between 2015 and 2019. The source apportionment based solely on the particle PNSD data was performed using PMF. A novel approach to analyse the data was used, as the particle PNSD data were combined from two nearby sites. As a result, the same factors with the same time series were obtained for both sites, only with different profiles. If a similar profile was seen at both sites the source was considered regional.

Five factors were found in the data: TRA1, TRA2, SCA, SecA, and LRT. Three of the factors were related to traffic. TRA1 had a clear diurnal profile, with the highest peak during the morning rush hour and the second, slightly lower peak during the afternoon rush hour. TRA2 peaked approximately 1 hour later than TRA1, indicating slight processing in the atmosphere. SCA reached a maximum 3 hours later than TRA1, being much more aged. SCA had an evening peak in addition to the morning rush hour peak, which indicated that it might have originated from both liquid fuel (mainly traffic) and solid fuel (biomass) combustion. TRA1 was the main contributor to PNC at the SC, while at the UB, the PNC was usually dominated by the slightly aged combustion-related factors TRA2 and SCA. During summertime, SecA also had a significant contribution to aerosol PNC at both stations.

The trend analysis revealed that TRA1 and SecA have been decreasing by 9.1% and 6.6% yearly, respectively. For TRA2, an increasing trend of 6.5% yearly was discovered. These findings indicate that the properties of particle emissions originating from traffic have changed in recent years, probably due to the changes in vehicle engines and after-treatment techniques. The significant decreasing trend for TRA1 implies that while the improved emission reduction techniques seem to be reducing freshly emitted particulate emissions of traffic, the slightly aged traffic emissions are even increasing, as an increasing trend was observed for TRA2. This change in vehicle fleets is not only related to direct emissions; decreased SecA can be speculated to be linked to decreased core particle concentrations, such as BC.

The SCA factor seemed to be a mix of aged traffic particles and particles from biomass combustion. However, the contribution of biomass combustion to the PNC in the traffic environment entails high uncertainty. Additionally, all the factors had more than one mode. Therefore, in addition to the particle size bins, adding the auxiliary data to the PMF analysis might improve the separation between the factors. However, the novel method of attaching simultaneous data from two sites seems to improve the detection of various factors and could be useful in locations where PNSD data is available from more than one site.

In conclusion, traffic remains a large contributor to ambient PNCin urban environments despite the decreasing trend caused by the improvements in emission reduction technologies and electrification of the traffic fleet. Additionally, while the primary

emissions have decreased the effect on the secondary aerosols is more uncertain; in this study, the concentrations of slightly aged aerosols were increasing. Therefore, studying how emissions age in the atmosphere is important in the future.

Additionally, the study demonstrated that detecting aerosol source factors purely based on PNSD data is possible, but attaching the factors to individual sources would be difficult without available auxiliary data.

## Data availability

Data is available upon request from the corresponding author Sami Harni (sami.harni@fmi.fi)

## Supplement

**Author contributions**

SDH created formal analysis, software, and visualization and wrote the original draft of the paper. MA, SS, and JN contributed to the conceptualization and writing by reviewing and editing the article. HP, PA, and HM contributed to the investigation. VL contributed to the formal analysis, reviewing, and editing of the article. PH contributed to the formal analysis. TP and TR contributed to editing and rewriting the article. HT acted as a supervisor and contributed to the conceptualization as well as

reviewing and editing of the article.

## Competing interests

One of the co-authors is a member of the editorial board of Atmospheric Chemistry and Physics.

## Acknowledgments

This work was supported by the European Union's Horizon Europe 2020 research and innovation programme under grant

agreement No 101096133 (PAREMPI: Particle emission prevention and impact: from real-world emissions of traffic to secondary PM of urban air), grant agreement  No 814978 (TUBE), and Grant agreement No 101036245 (RI-URBANS). Financial support came from the Urban Air Quality 2.0 project, funded by Technology Industries of Finland Centennial Foundation, and from the Black Carbon Footprint project, funded by Business Finland (Grant 528/31/2019) and participating companies. The work in Rochester, NY was funded by the New York State Energy Research and Development

Authority under contracts #59802 and 125993. AI tools were used to improve the language of the article.

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
