# Peer review of "Source apportionment of particle number size distribution at the street canyon and urban background sites"

_EGUsphere, 2023_

## Author Comment (AC1)

We thank the reviewer for taking the time to read the article and for valuable comments on our paper. To facilitate the revision process, we have copied the reviewer comments (in black text) and our responses are in blue font. We have responded to all the reviewer comments and made alterations to our paper (**in bold text**).

The authors have first numbered the comments of the reviewer one to help the answering process.

Reviewer #1: Harni et al. used positive matrix factorisation to explore the particle number size distributions simultaneously collected at two spatially adjacent urban sites. The dataset presented here is unique, as simultaneous particle size distribution measurements are still very rare. Using the well-established PMF methods to explore such a unique dataset is of high interest to the aerosol community. However, the current version of the manuscript is not well written. The way in which data are presented, and the use of language significantly affects the quality of the research. The manuscript needs major revision to enhance the overall quality. I would support the final publication after addressing my comments below and polishing the language.

**AR:** We thank the Reviewer for these positive comments. The language is now thoroughly checked *on www.proof-reading.com*.

Major Comments

1. Lines 36 – 45: The current summary of source apportionment approaches is not comprehensive enough. The authors are expected to provide a compact summary of various source apportionment approaches used in particle size distribution data. In addition, the authors should provide a rationale for why the PMF was chosen as the source apportionment method in this study.

**AR:** The authors acknowledge that the summary of the source apportionment methods was not comprehensive enough as it was therefore modified and the rationale for the usage of PMF was also added:

**Commonly used source apportionment techniques in atmospheric sciences include k-means cluster analysis, principal component analysis (PCA), and receptor modelling methods. In this work, a receptor modelling method called positive matrix factorization (PMF) was used. PMF is a mathematical multi-derivative method developed by Paatero, (1997) that can be performed for many types of data, and it is the most widely used and established source apportion method for atmospheric aerosol particle data currently (Hopke et al, 2020; Hopke et al., 2022; Yang et al, 2020). The decision to use PMF was made because PMF is a well-established source apportionment method in environmental sciences, and there was suitable software available. Additionally, as PMF is a factor analysis method, it is fundamentally suitable to this kind of study, as it assumes that the observed data is a combination of latent underlying factors. In contrast, PCA, for example, attempts to linearly combine the underlying variables to reduce the size of the data. PMF has been used for chemical composition data (e.g. Li et**

**al. 2003; Makkonen et al., 2023), mass spectra (e.g. (Oduber et al., 2021; Teinilä et al., 2022), particle number size distribution (NSD) (Krecl et al., 2008; Zhou et al., 2005) and in combined matrixes with NSD and auxiliary data (Rivas et al., 2020).**

2. Lines 42 – 44: Why is conducting source apportionment solely based on NSD data and using auxiliary data only challenging? If it is indeed challenging, why did the authors still choose to use source apportionment in the study? What is the novelty of this study? More explanations should be provided here.

AR: Authors agree that the reasoning behind taking this approach could be elaborated and therefore the text is edited to the following form:

**However, conducting source apportionment solely based on NSD data and using auxiliary data only to verify the sources seems to have some challenges as the source profiles might be mixed with multiple sources (Zhou et al., 2005, Jollife & Cadima, 2016, Krecl et al., 2008). This makes interpreting of results using auxiliary data more difficult. To improve the separation between sources when using only NSD data as input to PMF, NSD data from two sites is combined into one data file in this study.**

Additionally, the novelty of this study is now stated at the end of the last paragraph of the introduction:

**The novelty of this study is how the data was handled from the two nearby sites with strongly overlapping aerosol sources by adding the data from the two sites to the same data matrix as columns instead of doing two separate PMF analyses.**

3. Section 2.2: Have the two DMPS systems been compared against each other? If not, please comment on whether the different instrumentation could affect the PMF analysis.

AR: Both of the DMPS systems have been compared with the reference SMPS Tropospheric Research (Tropos) in the summer of 2021 and the results were found comparable.

[Figure]

*Figure 1: Intercomparison results of the system at the UB station with the reference instrument from Tropospheric Research (TOPOS)*

[Figure]

*Figure 2: Intercomparison results of the DMPS system at the SC station with the reference instrument from Tropospheric Research (TOPOS) for the 21st of June 2021.*

[Figure]

*Figure 3: Intercomparison results of the DMPS system at the SC station with the reference instrument from Tropospheric Research (TOPOS) for the 22nd of June 2021*

[Figure]

*Figure 4:Intercomparison results of the DMPS system at the SC station with the reference instrument from Tropospheric Research (TOPOS) for the 23rd of June 2021*

[Figure]

*Figure 5: Intercomparison results of the DMPS system at the SC station with the reference instrument from Tropospheric Research (TOPOS) for the 24th of June 2021*

AR: The intercomparison is now also mentioned in the manuscript:

**Both DMPS systems participated in an intercomparison with a reference instrument from Leibniz Institute for Tropospheric Research (TROPOS) SMPS in the UB station between June 11 and 14, 2021, and demonstrated comparable results.**

4. Lines 165 – 167: It is still unclear why the five-factor solution is the best solution in terms of mathematical and physical aspects. As a PMF user, I typically present the Q/Qexp values, residuals, relative residuals and scaled residuals of different PMF solutions, when analysing the PMF results of aerosol mass spectra data. Presenting these aspects will strengthen the statistical significance of the chosen PMF solution. In addition, why does the five-factor solution have the best physical meaning? To convince the readers, the authors are encouraged to present other factor solutions that neighbour the chosen PMF solution in the Supplement.

AR: We acknowledge that the selection of factor number is subjective and depends on the PMF user. We have added new validation figures and the neighboring solutions to the supplement. We have also added a discussion on the reason for the selected number of factors. The old Figure 2 presenting the residuals for each size class was removed from the manuscript and replaced with the following Figure where the residuals, scaled residuals, relative residuals, and $Q/Q_{exp}$ values have been presented for each number of factors between 2-10:

[Figure]

*Figure 6: Mean residuals (A), mean scaled residuals (B), mean relative residuals (C), and $Q/Q_{exp}$ values (D) for the different number of factors between 2 and 10. The mean residuals presented have been calculated size-wise as an average over the unified dataset from the SC and UB measurement locations.*

Additionally, the text has been edited to the following form:

**Fig. 2 shows the mean residuals, mean scaled residuals, mean relative residuals, and $Q/Q_{exp}$ values for the different number of factors between 2 and 10. At the chosen five-factor solution, the mean relative residual was only around 2.8% on average. The residual and $Q/Q_{exp}$ values decrease continuously as the number of factors increases. However, for scaled residuals, mean relative residuals and $Q/Q_{exp}$ values, the decrease is smaller after increasing the number of factors past five. This is an indication that five is an acceptable number of factors. Additionally, the neighbouring solutions of four and six are presented in supplemental Figures S3 and S4, respectively. The four-factor solution merges the factors described later in this paper (SCA and SecA). In a five-factor solution, these two have notably different diurnal profiles, and therefore, merging them is not sensible. The six-factor solution presented in S3 splits the SCA into two factors that have very similar diurnal profiles and contributions throughout the year, and therefore, they are likely to be from the same source.**

The following figures of the four and six-factor solutions have been added to the Supplement:

[Figure]

*Figure S7: Four-factor solution of positive matrix factorization (PMF) factors presented for both stations on linear (A for SC, C for UB) and logarithmic x-axis. E presents the hourly relative contributions during workdays, F during weekends, and G the average monthly contributions. Note that the linear scale for plots A and B is different. The value presented in contribution figures is the factor with which to multiply the factor profile at any current time to get the total contribution. The average for the contribution factor is 1 over the whole measurement period for all the factors.*

[Figure]

*Figure S8: Six-factor solution of positive matrix factorization (PMF) factors presented for both stations on linear (A for SC, C for UB) and logarithmic x-axis. E presents the hourly relative contributions during workdays, F during weekends, and G the average monthly contributions. Note that the linear scale for plots A and B is different. The value presented in contribution figures is the factor with which to multiply the factor profile at any current time to get the total contribution. The average for the contribution factor is 1 over the whole measurement period for all the factors.*

5. Lines 185 – 187: Based on S3 – S12, I found it difficult to recognize whether the trends calculated using the seasonal Theil-Sen estimator and the Theil-Sen estimator calculated from data without seasonal variability were almost identical. Can the authors specify the trends mentioned in Line 185 with proper cross reference? In addition, the authors are expected to provide a quantitative comparison between the two types of estimated trends, instead of subjective observations.

AR: The lines are now modified to the following form to now have a proper cross reference:

**Figures showing the trend decomposition for the factors and the fitted Theil-Sen estimators are presented in the supplemental material. The trend decompositions for TRA1, TRA2, SCA, SecA, and LRT are presented in S7, S9, S11, S13, and S15, respectively. Additionally, the fitted Theil-Sen estimators are presented for TRA1, TRA2, SCA, SecA, and LRT in S8, S10, S12, S14, and S16 respectively.**

Additionally, a reference to Table 5 where the numerical values for trends calculated using both of the methods are presented is added to the manuscript:

**Notably, the trends calculated using the seasonal Theil-Sen estimator and the Theil-Sen estimator calculated from data without seasonal variability were almost identical, increasing the confidence in using seasonal trend decomposition for the data (Table 5).**

6. Lines 209 – 211: The complex sentence and grammatical mistakes make it very difficult to understand the mathematical and physical meanings of the contributions presented in Fig 4. Please rephrase the sentence.

AR: The sentence is now replaced with:

**The contributions depicted in Fig. 4 represent the scaling factors applied to each source profile at specific times. For instance, if the contribution at a certain time is two, the corresponding source profile is scaled by a factor of two at that moment. The source profiles and their contributions are normalized such that the mean contribution from each source averages to one over the measurement period.**

7. Lines 234 – 244: I agree that TRA2 is a traffic-related factor. The author also claims that TRA2 is slightly aged and atmospherically processed. However, the paragraph lacks a description of these two features. It is unclear "(~ minutes to an hour)" mentioned in the sentence. How can the authors come up with this aging time scale? In addition, how did the cooling, dispersion, and mixing impact TRA2?

AR: The authors acknowledge that there is no clear evidence to say that the time scale is from minutes up to an hour, and therefore this is removed from the sentence. Additionally, the mention of effects of cooling, mixing, and dispersion are removed, because although these effects play an important role in the shape of the particle number size distribution when the exhaust emission exits the tailpipe, the effect on atmospheric aging should not be so significant. Additionally, the reasoning behind expecting the aged nature of factor TRA2 and how is now better explained in the text:

**TRA2 was interpreted as a slightly aged traffic-related factor. Atmospheric aging of aerosols is expected to increase the mode particle size of NSD due to the oxidation of gaseous volatile organic compounds (VOCs) into compounds with lower volatilities. These oxidized compounds then condense on existing particles, making them larger. Furthermore, smaller particles experience greater diffusion losses. Consequently, we can expect a shift in the mode particle size toward larger particles during aging. The aged nature of the factor was concluded as shown in Fig. 4e; the morning rush hour peak of TRA2 is observed 1 hour later compared to TRA1. TRA2 also has similar contributions at the SC and UB, and therefore, the TRA2 was considered to be slightly aged, as the road is further away (100 m) from the UB station. Additionally, the mode**

**particle size was larger for TRA2 compared to TRA1, with a maximum mode particle size of 16.2 nm at both stations.**

Minor Comments

8. It is very confusing how the authors define the degree of Pearson correlation coefficients shown in Table 3 and the corresponding sentences. What is "strong", "significant", or "weak" in the context of Pearson correlation coefficients?

AR: Authors agree that the concept of the Pearson correlation coefficient is not well explained in the text. Now an explanation that the Pearson correlation coefficient measures a linear relationship between the variables is added to the data:

**Overall, the linear relationship between the variables (Pearson correlation coefficient R) for TRA1 with BC (AE33 with 880 nm) and NO$_x$ were 0.76 and 0.85 at the SC, respectively. TRA1 had also a high correlation with NO$_2$ and NO at SC (Table 4).**

9. Please summarise any other studies that conducted PMF analysis on urban NSD data in the introduction.

AR: The authors have gone through the literature and found several articles that conducted PMF on number size distributions in either urban, suburban, urban background, or residential areas. Some articles may be still missing, but the 19 articles found should be enough to give an image of the state of the PMF studies on number size distributions in urban environments. A thorough summary of these articles is not written as it would be too long but the following text and reference to the articles have been made:

[revised manuscript text omitted]

10. This is not the 1[st] study on the data simultaneously collected from the Street Cayon and Urban Background Station in Helsinki. In the Introduction, please provide a summary of the main findings from the literature that focuses on the data collected from these two sites.

AR: The summary of articles that have used simultaneous data that is related to the data used in this study from both sites is added to the text:

In this study, particle NSD was investigated in urban background (UB) and street canyon (SC) sites in Helsinki, southern Finland. The simultaneous data from these two sites have been analysed in previous studies. Okuljar et al. (2021) investigated the relative contribution of traffic and atmospheric new particle formation to the concentration of sub-3 nm particles. They utilized PNC data between 1-800 nm and auxiliary data from the stations. They found that the particle concentrations in the SC were higher over the whole size range. Additionally, they associated particles in the size range of 1-25 nm with local sources at the UB and found particles in the size range of 1-100 nm to have a dominant contribution from local sources in the SC. Rivas et al., (2020) used data from both sites in a study that applied PMF on NSD data across four European cities. They identified five factors for both stations: nucleation, fresh traffic, urban background, biogenic, and secondary.

Additionally, the following references have been added to the reference list:

Okuljar, M., Kuuluvainen, H., Kontkanen, J., Garmash, O., Olin, M., Niemi, J. V., Timonen, H., Kangasluoma, J., Tham, Y. J., Baalbaki, R., Sipilä, M., Salo, L., Lintusaari, H., Portin, H., Teinilä, K., Aurela, M., Dal Maso, M., Rönkkö, T., Petäjä, T., and Paasonen, P. Measurement report: The influence of traffic and new particle formation on the size distribution of 1–800 nm particles in Helsinki – a street canyon and an urban background station comparison, Atmos. Chem. Phys., 21, 9931–9953, https://doi.org/10.5194/acp-21-9931-2021, 2021.

11. Section 2.2: To improve readability, please include a table in the main text outlining the instruments and measured parameters.

AR: The table and the text referring to it are now added to the manuscript:

**The instruments used in the measurements are listed in Table 1**

Table 1: The list of instruments used in the measurements.

| Instrument | Station | Measured variable |
| --- | --- | --- |
| DMPS (CPC, A20 Airmodus, and Vienna-type DMA) | SC | NSD |
| DMPS (Twin-DMPS) | UB | NSD |
| Q-ACSM (Aerodyne Research Inc) | SC | Non-refractory $PM_1$ |
| TEOM (model 1405) | SC | $PM_{10}$ and $PM_{2.5}$ |
| Aethalometer (AE33, Magee Scientific) | SC | BC |
| APNA370 (Horiba) | SC | $NO_x$ |
| APOA-370 (Horiba) | SC | $O_3$ |
| LICOR (model LI-7000) | SC | $CO_2$ |
| APMA-360 (Horiba) | SC | CO |
| TEI42S | UB | $NO_x$ |
| TEI49 | UB | $O_3$ |
| APMA 370 (Horiba) | UB | $SO_2$ |
| APSA 360 (Horiba) | UB | CO |

12. Figure 1: For such long-term data, it is worthwhile to provide better statistics of the data. Please include the minimum, maximum and interquartile.

AR: Authors agree that the figure could be improved. Therefore, an upgraded figure also showing standard deviations, maximum, and minimum values calculated from the hourly averages is now added to the text:

[Figure]

*Figure 9: A presents monthly mean values for $I_{tot}$, B for RH, and C for T over the measurement period (2015-2019) in Helsinki. The constant blue line represents the monthly mean value. The bars show the standard deviation, and the red dots show the maximum and minimum values counted from hourly values.*

13. Lines 129 – 130: "The interpolation needs to be done on a logarithmic x-axis …", is very hard for general readers to understand. Could the authors provide visualized examples of interpolations on both logarithmic and linear x-axis in the supplement?

AR: The error caused by using the linear scale in the interpolations would not be easily observed by using the 16-size bins that resemble the size distributions at each site. To properly be able to show the underestimation of the interpolated value on the positive derivative of the size distribution curve and the overestimation of the interpolated value on the negative derivative of the size distribution, an example size distribution with fewer size bins was made. The Figure showing the effect was added to the supplement and the text referring to the figure to the manuscript:

[Figure]

*Figure S10: The figure illustrates the inaccuracies in the concentration when interpolating NSD without the logarithmic y-axis. Specially, it shows how concentrations can be underestimated when the derivative of the NSD curve is positive (Panel A), and overestimated when the derivative is negative (Panel C). Panel B provides a view of the entire NSD curve.*

The added text:

**This effect is demonstrated in supplement Fig. S1.**

> 14. Lines 159 – 160: The sentence is hard to follow. What was the set C3 in this study?

AR: The C3 was set at 0.1. The text was simplified to the following form:

**where $\sigma_{ij}$ is measurement uncertainty, $C_3$ is an arbitrary constant that was set in this study to 0.1 for both the UB and SC, and $N_{ij}$ is the concentration of bin j of sample i.**

> 15. Lines 168 – 169: What is the dispersion correction? Could the authors provide a detailed description of that and references?

AR: Authors agree that a brief description of a dispersion normalization in the manuscript would be justifiable therefore the text is modified to the following form and a reference to the article describing dispersion normalization is added to the text:

**The dispersion-corrected PMF results were also calculated for the five factors for comparison, and the difference in the results calculated without the dispersion correction was found to be negligible (Dai et al., 2021). The differences between the normal and dispersion normalised PMF were evaluated based on the Pearson correlation coefficients between workday diurnals (>0.98 for all factors), weekend diurnals (>0.98 for all factors), monthly contributions (>0.96 for all factors) and factor profiles (>0.97 for all factors). In dispersion correction, the original measurement data is normalized by the ventilation coefficient which is the height of the boundary layer times the average wind speed during the period. The goal of the dispersion correction is to reduce the inaccuracy in the source apportion caused by the dispersion of aerosol in the atmosphere (Dai et al., 2021).**

The reference added to the reference list:

**Dai, Q., Ding, J., Song, C., Liu, B., Bi, X., Wu, J., Zhang, Y., Feng, Y., & Hopke, P. K.: Changes in source contributions to particle number concentrations after the COVID-19 outbreak: Insights from a dispersion normalized PMF. Science of The Total Environment, 759, 143548. https://doi.org/10.1016/j.scitotenv.2020.143548, 2021.**

16. Figure 2: To understand the statistical robustness of the PMF solution for the long-term dataset, it is better to provide the histograms to visualise the relative residuals for all NSD at each size.

AR: The whole of Figure 2 was changed as a response to comment 4 of reviewer 1. Therefore Figure 2 now presents the means of residuals, scaled residuals, relative residuals, and $Q/Q_{exp}$ for all of the PMF solutions between 2-10 factors instead of a histogram showing the average relative residuals for each size bin of the 5-factor solution. The authors appreciate the comment of the reviewer, but the histograms are not made separately for each of the size bins at both stations as this would result in a figure that consists of 32 separate histograms and would be hard to interpret. However, a modified version of the original Figure 2 is added to the supplemental material showing the average relative residuals for each size bin at both of the stations. The variation of the relative residual is indicated in the figure showing the standard deviation with an error bar added to the supplemental material and text referring to it in the manuscript:

**In supplemental material S5, the average relative residuals with the standard deviation are presented for each size bin.**

[Figure]

*Figure S11: Relative residuals with standard deviations presented for each size bin at SC (A) and UB (B) sites.*

17. Figure 3: Please include the R-squared values as a function of particle size for each year in the lowest panel.

AR: The R-squared values are changed to Pearson correlation coefficients as a response to reviewer comment 23 of reviewer 1. The Pearson correlation coefficients have now been presented for the years 2015-2019 separately in the lower panel and the caption of the figure is changed accordingly. Additionally, the font size in the figure has been slightly increased and titles have been added to subplots as the reviewer suggested in one of the minor comments.

[Figure]

*Figure 12: Time series of daily average NSD for SC and UB for each year: 2015 (A and B), 2016 (C and D), 2017 (E and F), 2018 (G and H), and 2019 (I and J). The data used is reduced to 16-size bins. The particle diameter ($D_p$) is presented on the y-axis, the x-axis presents the time, and PNC ($cm^3$) is shown by logarithmic color scale. The yearly correlation between the UB and SC stations (Pearson correlation coefficient) is presented in the bottom plot for the various particle sizes and daily mean concentrations.*

18. Line 193: In the sentence "The observed NSD at SC…", what is the size range of the nanosize particles?

AR: The particle size range as well as the nanosize particle size range asked for in the next comment were added to the text. The new text is presented after the next comment.

19. Line 197: What is the size range of the nanosized particles?

AR: Now the particle size range is added to the text:

**The observed NSD at the SC in the size range of 12.6 to 562 nm contained significantly more nanosized (< 100 nm) particles on many occasions, as well as higher overall particle concentrations compared to the UB.**

20. Figure 4: For the bottom three contribution plots, please provide the maximum, minimum and interquartile ranges.

AR: This is a great comment but technically very challenging. The authors think that adding interquartile ranges, maximum values and minimum values to the plots where there are already five different graphs in each would make the figure incomprehensible. Additionally, the minimum value for all the factors is likely to be near zero for all the time periods and the maximum value is likely to be very high. Adding these to the figure would increase the y scale so much that the diurnal pattern would be impossible to see. Therefore, Figure 4 is left unchanged.

21. Line 224 – 225: I cannot see how the finding from Ronkko et al (2017) is associated with the TRA1 data here. If not relevant, please remove the sentence.

AR: The sentence is now removed.

22. Figure 5: It is unnecessary to separate the $NO_x$ and BC into two plots. Please combine them into one by introducing one extra Y axis.

AR: The number of panels in the figure has now been reduced from for to two by adding the third y-axis to the figures:

[Figure]

*Figure 13: Diurnal profiles for TRA1 factor-related PNC concentrations at the SC, along with $NO_x$ and $BC_{880}$ concentrations presented separately for workdays (A) and weekends (B).*

23. R-squared value vs Pearson Correlation Coefficient: I find that the R-squared value was used for data presented in Figure 3, while Pearson Correlation Coefficient was used for data presented in Figures 5 and 6 and Table 3. What is the difference between the R-squared value and the Pearson Correlation Coefficient? What is the right occasion to use one, not the other, and vice versa?

AR: The difference between R-squared and Pearson correlation coefficient is that R-squared is typically used for comparing models, and Pearson correlation coefficient for comparing linear correlations between variables. In the case of models, the R-squared represents the square correlations between observations and values predicted with the model. The model

explains the r-squared times 100% of the variance in the explained variable. Therefore the right variable to use in Figure 3 and the text overall is the Pearson correlation coefficient and it is now corrected to the text and Figure 3. The modified figure is presented already as a response to comment 17 from reviewer 1.

24. Line 240: Please clarify "a slightly larger area".

AR: Authors agree that the used term is not optimal and therefore the sentence is changed to the following form:

**TRA2 also has similar contributions at the SC and UB, and therefore, the TRA2 was considered to be slightly aged, as the road is further away (100 m) from the UB station.**

25. Lines 240 – 241: According to Table 3, the correlation of TRA2 with $NO_x$ and NO emissions at SC is also significant as well. Please comment on this.

AR: Authors agree that the TRA2 also correlates with $NO_x$ and NO concentrations at the SC. To note this in the article the following text explaining the correlation is added to the manuscript:

**Additionally, TRA2 correlated moderately with $NO_x$ and NO measured at the SC. This moderate correlation was expected, as the $NO_x$ and NO measured at the UB are the background levels that are also measured at the SC. However, in addition, the $NO_x$ and NO concentrations at the SC are strongly influenced by the immediate traffic emissions, and therefore, the correlation of TRA2 with the concentrations at the SC is lower than with the concentrations at the UB.**

26. Line 244: If the boundary layer was shallow, should we expect SCA and SecA to have similar monthly trends as TRA2 due to the accumulation of pollutants?

AR: This is a great comment but not so straightforward answer exists. Authors agree that the SCA and SecA should also be higher if the boundary layer would have a significant role in the higher contribution during winter. However, Seca contribution is strongly affected by sunlight and oxidant concentrations, and thus maximum in summer is expected. Whereas for the SCA two processes are affecting annual variation. Firstly in summer, the combustion is less frequent but the meteorological conditions would favor SOA formation. Secondly in winter, the combustion emissions are significantly larger however the amount of UV light in Nordic regions is minimal which likely affects secondary aerosol formation. We have modified this sentence to consider only primary pollutants:

**In addition, the boundary layer is shallower during cold months enhancing the accumulation of primary pollutants.**

27. Lines 246 – 247: It is unclear why the weak correlation between SCA and auxiliary data is expected for atmospherically processed aerosol. Please clarify this in the main text.

AR: We consider that SCA forms in the atmosphere during the aging process and is affected by existing anthropogenic and biogenic VOCs, oxidant concentrations as well as meteorological conditions. Thus, we expect that the correlation with the primary exhaust species would be lower. However, the sentence was modified to refer to primary traffic emissions rather than all auxiliary data as it should have been:

**SCA had relatively weak correlations with the primary traffic emissions (e.g., NOₓ, BC, CO, ORG57) data, as could be expected for atmospherically processed aerosol.**

> 28. Figure 6: It is unnecessary to separate the NOₓ and m/z 60 into two plots. Please combine them into one by introducing one extra Y axis.

AR: The number of panels in the figure has now been reduced from for to two by adding the third y-axis to the figures:

[Figure]

*Figure 14: Diurnal profiles for SCA factor-related PNC concentrations at the SC, along with NOₓ concentration at the UB station and organic fragments at m/z 60 concentrations at the SC during workdays (A) and weekends (B).*

> 29. Line 295: Why did SecA have a high contribution during summertime?

AR: Authors think that SecA has a high contribution during summertime as the higher radiance leads to more oxidation and therefore more secondary aerosol production. This should be seen as an increase in the contribution of organic aerosol mass. A reference showing the increase of organics is now added to the text.:

**Barreira et al. (2021) described the increased contribution of organics aerosol mass during summertime in Helsinki.**

> 30. Line 304: Different from what?

AR: The sentence is now corrected to the following form:

**Notably, the contributions of the various factors to volume concentrations were different when compared to contributions to the PNC (Fig. 8).**

> 31. Line 328: What is meaning of "development of technology"? Please clarify the sentence.

AR: The meaning of "development of technology" is now clarified and the sentence is modified to the following form:

**The latter is closely connected to relatively stable biogenic sources and meteorology (T and I_tot) and the former to the fast development of cleaner engine and after-treatment technologies driven by new emission limits.**

32. Lines 331 – 332: "More data (years)… the better they hypothetical test is" is unclear. Is the decreasing trend for this study or Luoma et al., 2021?

AR: This sentence was referring to this study and is now modified to the following form:

**We note that more data (years) is needed to see if the trends found in this study were real as the larger the sample size is, the better the hypothetical test is. Table 5 presents all the results of the trend analysis.**

33. Table 4: What does "no significance" mean? Please add discussion in the paragraph associated with Table 4.

AR: Now the significance is mentioned in the caption of Table 5 as follows:

**The trend in the table is significant if the significance level is above 95% (p-value < 0.05).**

Additionally, the discussion about SCA and LRT was added to the text:

**No statistically significant trend was found for SCA or LRT, although a slight decrease was indicated for both. There could be many reasons for this. Although traffic emissions have decreased (e.g. Barreira et al., 2021), biomass combustion for residential heating has increased lately. LRT emissions are mainly affected by meteorological conditions and can vary a lot between years.**

34. Line 356: What does "immediate emissions of traffic" mean?

AR: The sentence was clarified to the following form:

**The significant decreasing trend for TRA1 implies that while the improved emission reduction techniques seem to be reducing freshly emitted particulate emissions of traffic, the slightly aged traffic emissions are even increasing, as an increasing trend was observed for TRA2.**

Technical Comment

A bunch of sentences are very hard for the readers to understand due to grammatical mistakes. I would encourage the authors to find professional language services or native speakers to polish the language. E.g.,

35. Lines 15 – 16: "This study aims to … sources using a novel approach to positive matrix factorisation (PMF)"

AR: The sentence is now modified to the following form:

**This is intended to improve the understanding of urban aerosol sources, utilizing a novel approach to positive matrix factorization (PMF).**

36. Line 235 – 237: "The mode particle size was … growing the particle size (Ning & Sioutas, 2010) "

AR: The sentence is removed from the manuscript as a part of answering comment 7 from reviewer 1.

37. Line 248 – 249: "The strongest Pearson correlation ... during both workdays and weekends"

AR: The sentence is now modified to the following form:

**The strongest Pearson correlation coefficient of 0.56 was observed between TRA2 and NO$_x$ at the UB site (Table 4). TRA2 and NO$_x$ at the UB site also had similar diurnal patterns on working days and weekends.**

38. Lines 346 – 347: "TRA2 peaked approximately 1 h later … being much more aged"

AR: The sentence was modified to the following form:

**TRA2 peaked approximately 1 hour later than TRA1, indicating slight processing in the atmosphere. SCA reached a maximum 3 hours later than TRA1, being much more aged.**

39. Lines 359 – 360: "Although the contribution from biomass combustion in traffic environment includes high uncertainty".

AR: The sentence is now modified to be more clear:

**However, the contribution of biomass combustion to the PNC in the traffic environment entails high uncertainty.**

40. Provide labels (e.g., (a), (b), (c)) for each subfigure in the main text and SI. To enhance the readability, please correct the cross-references throughout the whole text.

AR: This has now been done.

41. Lines 18 – 19: "The data is combined into one file so that the data from both stations has the same timestamps. Then PMF finds profiles for the unified data" are too descriptive. They are better to be placed in section 2, Experimental, instead of Abstract.

AR: Authors agree that this might be too specific for the abstract. However, it is necessary to say something about how the PMF analysis is done also in the abstract and therefore the text is edited to the following form:

**The particle source profiles were detected in particle number size distribution data measured simultaneously in a street canyon and at a nearby urban background station between February 2015 and June 2019 in Helsinki, southern Finland. The novelty of the method is combining the data from both sites and finding profiles for the unified data.**

42. Lines 30 – 31: What are the concentrations in the sentence "Of these, anthropogenic… concentrations." Are they number or mass concentrations?

AR: The sentence was misleading as it was meant to emphasize that anthropogenic sources are the dominant source in urban environments. Therefore the sentence is now modified to the following form:

**Of these anthropogenic sources are predominant in urban areas (Guerreiro et al., 2015).**

43. Lines 32 – 33: Please specify what are the indirect financial consequences.

AR: The financial consequences are now written open:

**The negative health effects related to particulate matter (PM) pollution (PM$_{2.5}$ and PM$_{10}$) are commonly accepted and well-documented (i.e., Koenig, 2000; J. Wu et al., 2017), also leading to indirect financial consequences through increased mortality and treatment of respiratory and cardiovascular diseases (Johnston et al., 2021).**

44. Line 82: Instead of using the smallest particles here, please provide the particle size range where the charging difficulties occurred.

AR: The sentence is modified to include the particle sizes of the problematic sizes:

**The DMPS charger had difficulties charging the three smallest particle size bins (6.0, 7.3, and 9.0 nm) on the SC site; therefore, particles smaller than 10 nm were excluded from the analysis for both sites.**

45. Lines 107 – 108: Provide reference(s) for "These relatively relative criteria for … in the traffic environments."

AR: The reference was not added to the sentence but the sentence was modified to better explain the reasoning behind the relaxed outlier criteria meanwhile making the the reference not needed:

**These relatively relaxed outlier criteria were applied because the measurement site is less than a meter from the driving lanes, and therefore, variations in the concentrations can be expected due to passing cars.**

46. Lines 117 – 118: "So the factor profile … between the stations." The sentence is too long to read. Please rephrase the sentence.

AR: Authors agree that the sentence is hard to follow and therefore the sentence and also some of the text around that sentence is now edited to the following form:

**In this approach, a single common factor is calculated for both stations, comprising 32 size bins. The initial 16 size bins are associated with the SC and the remaining 16 with the UB. Given that there is only one set of factors, the time series are identical for both stations, whereas the size distribution profiles vary between the sites. If a factor has a substantial local contribution at one of the sites but not at the other, then its profile would be pronounced at that station and near zero at the other.**

47. Figure 3: Please increase the font size and clearly label the year for each subfigure.

AR: OK, done.

48. Line 241: Is it supposed to be Table 3?

AR: Yes it should be. This is now corrected.

49. Line 282: Add a space for Barreiraet al., 2021.

AR: The citation is now corrected.

48.Table 4: Is the unit supposed to be "cm$^{-3}$"

AR: Yes, it should be. This is now corrected.

Reviewer 2 #The manuscript under review utilizes a more than valuable dataset from a northern European urban area, centered around long-term PNSD measurements from two stations of different characteristics (urban background vs street canyon). These data are complemented by various equally valuable measurements and can be characterized as a unique dataset that should be analyzed and published. However, the presented analysis lacks depth and focus. Is the method proposed or the results/findings that are important? After some necessary improvement and quite some language editing, the article can be suited for publication.

**General Comments**

1. The introduction can definitely be improved, in terms of providing the context and pinpointing the gaps that this study aims to fill, but especially when the authors are setting the scientific goals of the study, that are now only presented in one sentence (**Lines 58-59**). Furthermore, this very stated goal, i.e. "to explore how well the sources of urban aerosols are statistically separable based on the number size distribution data using positive matrix factorization", does not seem to be addressed in the main text. How did the authors explore the source separating power of PMF on PNSDs. Did they compare with other approaches (e.g. clustering)? Did they compare with PMF that also included other variables? Furthermore, what were the statistical tools used to asses the above?

AR: The introduction is already heavily modified as a response to the comments of the first reviewer. The aim was poorly defined. Our aim was to find ways to better use commonly measured size distribution data. The tool (EPA PMF 5.0) is developed by EPA and described elsewhere. We tested different approaches for the data analysis (with and without additional data, analyzing stations separately, and different numbers of factors). However, finally, we found this chosen approach to be most suitable for this data set and our scientific goals. We have now added information about the validation of the chosen solution. We agree that in the ideal case, one would analyze the dataset with multiple different tools and compare the results. Analyzing this large data set even with one tool was really time-consuming and challenging and thus it was not feasible to use several different approaches. The aim was reformulated to better represent the aims of this study:

**This study was intended to improve the understanding of urban aerosol sources by applying statistical source apportionment methods such as PMF (EPA PMF 5) to long-term size distribution data.**

2. The authors should try to provide more details on why they selected to combine the two datasets in a single PMF input by concatenating the data matrices horizontally? What was the scientific goal behind such a choice? How was the PMF result, i.e. a single timeseries for both stations advantageous in characterizing the sources affecting the sites? Why not perform PMF in each station separately? Or for instance, why not perform a standard multi-site PMF (Pandolfi et al., 2020; van Pinxteren et al., 2016)

given that the authors state in **Line 117** that "the same factors were solved for both sites"?

AR: The authors were not able to do the analysis separately for the stations. The results were not satisfactory. We tested many different approaches and found this to be the best solution. EPA PMF 5.0 is a well-described method and s freely available open-access tool which was considered beneficial. The sites are located only less than a kilometer apart, thus the sources are bound to be fairly similar. The only major difference is the distance to the major road (SC 1 m, UB > 100 m) thus analyzing the dataset together provided a possibility to study the influence of traffic specifically. In this study, the standard multi-site Positive Matrix Factorization (PMF) was not utilized for analysis. The reason is, that it presumes that all measurement sites share an identical set of sources. However, this assumption does not hold true for our case, as our study involves two distinct sites: a traffic site and an urban background station. Our focus is primarily on the source apportionment of the number size distribution, rather than the chemical composition of particles, a topic explored by Pandolfi et al., (2020), for instance. This focus introduces two significant factors that result in the sources at both sites not being identical. Firstly, the traffic site is characterized by fresh emissions from vehicles. Secondly, the Number Size Distribution (NSD) of these fresh emissions does not remain constant as the emissions disperse further from the road. This is in contrast to the chemical composition, which remains approximately the same. Given these factors, it was not anticipated that the fresh traffic NSD would be detected at the urban background site. Thus, the standard multi-site PMF could not be applied effectively in this context.

3. In order to justify applying PMF this way, a comparison with a standard PMF on each site separately, as well as with a standard multi-site PMF should be added and the performance of the proposed method should be assessed in a quantitative way. In general, the method description is not verry straightforward. A step-by-step schematic of the measurement matrix manipulation before feeding PMF should be added in the main text or supplement.

AR: The authors argue that doing a full comparison with horizontally combined datasets to doing standard PMF for both stations separately would be an interesting comparison and would be an interesting article on its own. However, doing the comparison as a part of this article would deviate too far from the scope of this study and is not therefore done in this study. Comparing the results to the multi-site PMF would not be reasonable because of the reasons presented in the previous answer to comment 3 from reviewer 2.

The method description is already partially improved as an answer to comment 46 of reviewer 1. In addition, a step-by-step schematic of the data matrix manipulation before feeding the data to PMF is now added to the manuscript, and the text is modified to the following form:

**The data was processed in the following manner before being input into the PMF: Initially, outliers were identified and eliminated separately at both stations. Subsequently, the data was averaged on an hourly basis independently at each station. The data was then interpolated to 16-size bins at both locations. Finally, the data from the two sites was merged horizontally into a single matrix with 32 bins in total. In more detail, strong outliers were removed from the DMPS data by calculating the total concentration and removing the data points that had a concentration ten times larger or smaller than the adjacent measurement points.**

4. I believe that the reasoning behind the selection of the presented 5-factor solution is poorly presented in the manuscript. To my opinion more quality metrics should be presented (e.g. Q/Qexp, temporal trends in residuals etc). Such analysis should be presented in the context of comparison to more and less factor solutions. For instance, how do such metrics change when moving from a 6-factor to a 5-factor and then to a 4-factor solution. The robustness of the solution should be examined and presented. Was bootstrap resampling or displacement analysis performed? Is the solution repeatable among different runs starting from different random seeds? How was rotational ambiguity addressed?

AR: The validation metrics were added to the manuscript (see answer 4 to reviewer 1). Also, the comparison to 4 and 6-factor solutions was added to the supplement, and a discussion related to them to the manuscript. See answer to the comment 4 of reviewer 1. The most critical displacement analysis was considered more important and the Bootstrap method suffers from technical limitations in EPA PMF 5.0 (software crashes repeatably when Bootstrap is run). The displacement error analysis was run on the 5-factor solution and the results showed little to no rotational ambiguity. Sadly bootstrapping could not be done in EPA PMF 5.0 because of the large size of the data. The solution was also always very repeatable when the random starting seeds were tested. The following text is added to the manuscript:

**The robustness of the solution was tested using different random seeds as starting points and performing displacement analysis on the solutions.**

5. The discussion regarding the SCA factor needs more detail, given that this factor is presented to include various types of combustion related aerosol. What seems odd is that, while it is stated that it also represents biomass burning contributions, its average contribution does not exhibit a wintertime enhancement. Don't the authors expect a more pronounced contribution of biomass burning at the UB site? Is that reflected in the results?

AR: The wintertime enhancement is not seen as while during the winter time the amount of biomass combustion increases the amount of sunlight and available oxidants is small limiting the amount of SOA formation, whereas during summer the amount of biomass burning is lower but the amount of oxidants and sunlight increases. The distance between the sites is less than one kilometer and the UB station is surrounded by the university campus and botanical garden. Previous source apportionment analyses have shown that the contribution is small also in UB e.g. according to Timonen et al., 2013 the contribution of local biomass combustion in the UB site was 4 % to organic aerosol mass. The following discussion about the annual variation is added to the manuscript:

**The annual variation of SCA is small (Fig. 4g). Likely because although during the wintertime, the amount of biomass combustion increases the amount of sunlight is low, limiting SOA formation, whereas during summer, the amount of biomass burning is lower, but the amount of sunlight increases, thus enhancing SOA formation. In contrast, traffic emissions remain stable throughout the whole year.**

6. Wind regression analysis would greatly help the interpretation of the results and add to the quality of the presented analysis.

AR: We tested wind regression analysis but could not find additional information. In the SC site, the wind directions are heavily impacted by the nearby buildings which would make the wind regression difficult. Additionally, the major particle source at the SC site is likely to be the traffic that contributed to the aerosol concentrations regardless of the wind direction. Also, the UB site is located inside the city area and therefore traffic traffic-related aerosol would be transported to the site despite wind direction. The same is true for all other sources detected in this study. To use wind as help for the identification of long-range transport a simple wind regression analysis would not be enough but full-back trajectories would need to be used.

**Specific Comments**

1. **Lines 15-16**: See general comment #2.

AR: The text has already been modified as an answer to comment 35 from reviewer 1.

2. **Lines 18-19**: The sentences "The data is combined… for the unified data." Are not that well written and generally don't seem to belong to the abstract. In general, there should be some more effort for the abstract to capture the highlights of this research adequately. For instance, there is no single word on the trend analysis and its implications.

AR: The sentence that is referred to in the comment is already removed from the abstract novelty of this study and is stated more clearly as a response to reviewer comment 41 from reviewer 1. Additionally, the trends are now mentioned in the abstract:

**Additionally, the trends of the found factors were studied, and statistically significant decreasing trends were found for TRA1 and SecA. A statistically significant increasing trend was found for TRA2.**

3. **Lines 75-83**: Have the two instruments been intercompared? More details (manufacturer etc) should be provided for the UB DMPS. Some details or references on measurement quality should be provided. Were the sample streams dried in both instruments?

AR: The instruments have participated in intercomparison and relevant intercomparison results and reference figures on measurement quality have been presented as a response to the comments of 3 reviewer 1. Both instruments were manufactured at the University of Helsinki and were calibrated against reference in European Center for Aerosol Calibration and Characterization (ECAC) and operated with driers. This is now also added to the manuscript:

**Both of the DMPS systems were made by the University of Helsinki and approved by European Center for Aerosol Calibration and Characterization. Both of the systems had dryers in the inlet lines to keep the relative humidity (RH) below 40%.**

4. **Line 85**: More detail on the operation of the ACSM or relevant references should be provided. Was it calibrated? Was the sample dried? Were the data quality controlled? Was a chemical composition dependent collection efficiency used?

AR: The additional information considering the operation of the ACSM including information about calibration sample drying and quality control of the data and chemical composition efficiency. wit relevant references is now provided in the manuscript: following

reference to the manuscript showing a detailed description of the Q-ACSM measurement setting:

**A mass-based Q-ACSM calibration was performed using dried size-selected (300 nm of mobility diameter) ammonium nitrate and ammonium sulphate aerosol particles. The effective nitrate-response factor (RFNO3) relative ionization efficiencies of sulphate and ammonium (RIENH4) and relative ionization efficiency of sulphate (RIESO4) were determined, and analyte signals were converted into nitrate-equivalent mass concentrations. An effusive source of naphthalene, located in the detection region, was used as a reference for m/z and ion transmission calibrations. A Nafion dryer was installed prior to the instrument inlet so that the RH of the sample flow was maintained below 40%. A chemical composition-dependent collection efficiency was used having been calculated according to Middlebrook et al. (2012), with the exception that a collection efficiency of 0.45 was used for samples when ammonium was below the detection limit. More information can be found in Barreira et al. (2021).**

Also the following reference was added to the reference list:

**Middlebrook, A.M., Bahreini, R., Jimenez J.L., and Canagaratna, M.R.: Evaluation of Composition-Dependent Collection Efficiencies for the Aerodyne Aerosol Mass Spectrometer using Field Data. Aerosol Sci.Technol., 46:3, 258-271, DOI: 10.1080/02786826.2011.620041, 2012.**

5.  **Lines 136-138**: Couldn't the fact that the dataset was treated to its entirety, be a source of uncertainty, given that there can be substantial seasonal variability in observed sources? Did the authors perform any seasonal tests and establish that the number of factors remain invariable from season to season, or that their profiles were comparable to the whole period PMF?

**AR**: Authors agree that treating the data in its entirety might be cause for some uncertainty as the source profiles might vary slightly between the seasons. However, treating the data in its entirety was chosen as the source profiles were not expected to vary significantly between the seasons, and doing the PMF in one set allowed the authors to better investigate the possible trends of the different factors. This would have been more difficult if the PMF was done seasonally. The authors also experimented on doing the PMF on the data seasonally but it did not give any additional information. Additionally, all the found sources are expected to be found during all the seasons.

6.  **Lines 168-169**: A quantitative metric of the authors' choice, describing the comparison with the dispersion corrected PMF, should be provided. Furthermore, details on how the input data were treated for this dispersion corrected PMF run, how was the solution presented selected, along with relevant references should be added.

**AR**: The details of how the input data was treated and the relevant references are added to the text when answering comment 15 of reviewer 1. The solution (5-factor) presented for dispersion correction was chosen just because the best solution for the PMF without dispersion correction was 5 factor and the authors wanted to see if doing a dispersion comparison would cause any noticeable differences. The quantitative matric that was used in the comparison is now provided in the text:

**The differences between the normal and dispersion normalised PMF were evaluated based on the Pearson correlation coefficients between workday diurnals (>0.98 for all factors), weekend diurnals (>0.98 for all factors), monthly contributions (>0.96 for all factors), and factor profiles (>0.97 for all factors).**

7. **Line 180**: Please correct the typos, "Then-Seil" at the beginning of the sentence and "Thei-Sen" further on.

AR: The typos were corrected with an additional corrected typo in in Table 5.

8. **Line 240**: What about the afternoon? TRA1 exhibits a peak during the afternoon rush hour. Given that TRA2 is a product of primary traffic particles' atmospheric processing, shouldn't a TRA2 afternoon peak be expected? Is all the processing assumed to be linked with photochemistry? In that case would there be a summer-winter difference?

AR: The TRA1 follows more closely the amount of traffic as it represents the emissions measured from the immediate proximity to the measurement site. TRA2 on the other hand resembles the Slightly aged TRA2 emissions. During the day the boundary leyear height typically rises and therefore during the afternoon the traffic emissions have more air volume to disperse into therefore lowering the TRA2 concentrations. All of the processing is not assumed to be related to the photochemistry but also condensation and coagulation are likely to have an effect. If the photochemistry would be the dominant mechanism in the processing of TRA2 then the higher concentrations would be expected during summer time. The description of TRA2 processing has also been improved in the text as a response to comment 7 of reviewer 1.

9. **Line 244**: If BL dynamics could be part of the explanation for such pronounced seasonal variability for TRA2, shouldn't it also affect the other traffic related factor (TRA1)?

AR: The TRA1 factor is seen only at the SC site where the measurement site resides less than 1 m from the driving lanes. The TRA1 factor also represents the emissions measured from the immediate proximity of the measurement station. Therefore the height of the boundary layer is not likely to have significant effects on the TRA1 concentrations as the particles are emitted starting around 1 m from the sampling line and they do not have time to mix with the background to a degree that the boundary layer would have an effect before they are measured.

10. **Line 259**: This suggestion, that m/z 60 OA might be related to traffic, needs a citation or some more analysis to back it up. The diurnal pattern especially when not season specific, cannot support something like that on its own.

AR: Authors agree that the suggestion that m/z 60 OA is related to traffic is not well enough justified therefore the sentence is modified to the following form:

**Interestingly, m/z 60 was also elevated during the morning rush hour, although the m/z 60 is usually related to biomass combustion and not traffic.**

11. **Line 268**: How come the SecA factor correlates with $ORG_{43}$, that can be considered a primary OA marker? Could it be that such correlation is driven by specific reoccurring

events during wintertime? What about ORG$_{44}$, is there some correlation there? Could it be that the SecA factor has different origins when moving from wintertime to summertime? Does the selected PMF method allow such an assessment?

AR: ACSM is capable of measuring only unit mass resolution and cannot separate components that have the same mass-to-charge ratio. Org43 could be originated from e.g. hydrocarbons (e.g. C3H7+) or oxygenated carbon (e.g. C2H3O+). The former mainly originated from primary sources e.g. traffic and the latter one from oxygenated organic aerosol (secondary OA). Examples of high-resolution mass spectra from primary and secondary factors are presented for example in Saarikoski et al. (2021). Typically, less oxidized secondary organic aerosol factor (LO-OOA) has the contribution of Org43 to total Org (f43) higher than for more oxidized organic aerosol factor (MO-OOA) like long-range transported (Chen et al., 2022). Typically, sources or origins for LO-OOA are e.g. aged biomass burning in wintertime or biogenic organic aerosol originating from VOC in spring and summertime (Canonaco et al., 2015). However, it is also possible that SecA could have somewhat different origins when moving from summer to wintertime. The PMF analysis chosen does not allow an investigation of this. The following text is added to the manuscript:

**The m/z 43 has been associated with less oxidated secondary organic aerosol (Chen et al., 2022).**

And the corresponding reference:

Chen, G., Canonaco, F., Tobler, A., Aas, W., Alastuey, A., Allan, J., Atabakhsh, S., Aurela, M., Baltensperger, U., Bougiatioti, A., De Brito, J.F., Ceburnis, D., Chazeau, B., Chebaicheb, H., Daellenbach, K.R., Ehn, M., El Haddad, I., Eleftheriadis, K., Favez, O., Flentje, H., Font, A., Fossum, K., Freney, E., Gini, M., Green, D.C., Heikkinen, L., Herrmann, H., Kalogridis, A-C., Keernik, H., Lhotka, R., Lin, C., Lunder, C., Maasikmets, M., Manousakas, M.I., Marchand, N., Marin, C., Marmureanu, L., Mihalopoulos, N., Močnik, G., Nęcki, J., O'Dowd, C., Ovadnevaite, J., Peter, T., Petit, J-E., Pikridas, M., Platt, S.M., Pokorná, P., Poulain, L., Priestman, M., Riffault, V., Rinaldi, M., Różański, K., Schwarz, J., Sciare, J., Simon, L., Skiba, A., Slowik, J.G., Sosedova, Y., Stavroulas, I., Styszko, K., Teinemaa, E., Timonen, H., Tremper, A., Vasilescu, J., Via, M., Vodička, P., Wiedensohler, A., Zografou, O., Minguillón, M.C., Prévôt, A.S.H.: European aerosol phenomenology − 8: Harmonised source apportionment of organic aerosol using 22 Year-long ACSM/AMS datasets, Environ. Int, 166, 107325, https://doi.org/10.1016/j.envint.2022.107325, 2022.

12. **Lines 270-271**: This statement seems a bit speculative since the authors don't provide any information on the BC mixing state. How much of the BC measured is fresh at the SC site and how much at the UB. How long would it take for a shell to form? Is there a consistent time shift between concentration peaks in BC and SecA?

AR: Authors note that this is a valuable comment from the reviewer but the mixing state of the BC was not studied and therefore assuming how much of the BC is fresh would be highly speculative and is not done. However, during atmospheric aging, the VOCs condense on existing particles that can be solid BC particles existing in the car exhaust. The text is now modified into a form that necessarily does not indicate BC acting as a core but also being mixed into SecA particles:

**Surprisingly, the SecA factor also somewhat correlates with BC (Table 4), possibly indicating that BC particles that are ubiquitous in traffic environments might act as cores for or be mixed into these SecA particles.**

13. **Lines 273-283**: Any comment on the LRT factor seasonal variability, where an enhancement during the Jan – Mar period is observed? Moreover, in **Table 3** the highest correlation calculated for the organics concentration at m/z 60 was with the LRT factor. Did the authors exclude somehow that there is some factor mixing here?

AR: The seasonality of the LRT factor was not further studied. However, the LRT typically has events of different strengths that can last several days increasing the local PM concentrations in the Helsinki area significantly. During the relatively short period of this study of only 4 and half years, it is possible that some stronger LRT events have happened during January, February, and March of some years increasing the contribution of LRT during these months significantly. The somewhat high correlation of the LRT factor with m/z 60 is somewhat expected as biomass combustion and wildfires in eastern Europe are a significant source of LRT events in the Helsinki area and m/z 60 is a commonly used marker for wood combustion. However, it is possible that LRT also has some mixing with the regional background aerosols that could have a higher contribution from wood combustion as many houses and saunas outside the densely built Helsinki central area use wood combustion for heating.

14. **Lines 326-327:** I don't understand what the authors mean with these two sentences. Please rephrase to make the message clear.

AR: The latter of the sentences is already improved and modified as a response to comment 31 from reviewer 1. Now additionally the first sentence is changed to the following form:

**The decrease of the SecA factor is more complex, to explain as it was speculated to have both anthropogenic and biogenic sources.**

15. **Lines 328-329**: I believe that the authors here use a previous speculation as a fact in order to develop their argument based on decreasing BC concentrations. In fact, a "somewhat" correlation can't act as a solid basis for such a discussion. Please rephrase.

AR: Authors agree with the reviewer's comment that the somewhat correlation with the the BC is not a solid enough basis for such a discussion and therefore the sentence is modified to the following form:

**In the previous chapter, SecA concentrations were shown to indicate possible correlation with BC. One possible reason for this was suggested to be that the BC particles could act as cores for SecA.**

**Figures**

- **Figure 1:** Not so sure that this figure is necessary in the main text. It could be moved to the supplement.

AR: Authors considered moving Figure 1 to the supplement. However, the figure is later referred to in the text when the factors are identified, and it therefore is needed in the main text.

- **Figure 4**: Tick marks every 6 or 4 hours would greatly improve the readability of the diurnal contributions presented.

AR: This is now done:

[Figure]

*Figure 15: Positive matrix factorization (PMF) factors presented for both stations on linear (A for SC, C for UB) and logarithmic x-axis. E presents the hourly relative contributions during workdays, F during weekends, and G the average monthly contributions. Note that the linear scale for plots A and B is different. The value presented in contribution figures is the factor with which to multiply the factor profile at any current time to get the total contribution. The average for the contribution factor is 1 over the whole measurement period for all the factors.*

- **Figures 7 & 8**: These two figures are not actually discussed in the text. The relevant discussion could apply to some pie charts alone. More discussion should be provided in case the authors believe that the monthly variation reveals important information.

AR: Authors argue that presenting the contributions in the form of time series gives a better understanding of the trends discussed in the article. Using a pie plot representing the evolution of the contributions over the years would be difficult. However, the authors agree that more discussion on the monthly variation of the concentrations could be added to the manuscript. Some discussion related to the monthly variation seen in Figures 7 and 8 is now added to the text:

**The contributions of LRT to volume concentration varied greatly from month to month at both stations. The months of the highest concentrations varied between years, highlighting the event nature of this factor, as singular strong events can increase LRT contributions. This is in contrast to Fig. 7 and Fig. 8, which show little month-to-month variation, and the concentration patterns stay relatively stable between years.**

---

## Author Response (AR2)

We thank the reviewer for taking the time to read the article and for valuable comments on our paper. To facilitate the revision process, we have copied the reviewer comments (in black text) and our responses are in blue font. We have responded to all the reviewer comments and made alterations to our paper (**in bold text**).

Reviewer 1 # The quality and language of the manuscript have been significantly improved. However, the way in which the ACSM data is used to interpret different PMF factors still needs revision. I would recommend the manuscript be published once the following comments have been addressed.

Major Comments

1. If the dataset measured from the two sites has been analysed using PMF by Rivas et al. (2020), what will be the novelty of this study? Please provide a short statement in the Introduction.

AR: The end of the introduction chapter has now been modified to the following form to state the novelty of this study better although Rivas et al. (2020) have used partially the same data.

**The PMF has been applied to the data from these two sites earlier in a study by Rivas et al., (2020). However, they used data from January 2007 to December 2016 whereas data in this study is from January 2015 to June 2019. Additionally, Rivas et al., (2020) used NO$_2$, NO, SO$_2$, CO, and O$_3$ data in addition to PNSD data in the PMF input files to separate the sources. In contrast, in this study only the PNSD data was included in the PMF input files and the results were later compared to the other measurement data. In addition to this, the novelty of this study arises from how the data was handled from the two nearby sites with strongly overlapping aerosol sources by adding the data from the two sites to the same data matrix horizontally as columns instead of doing two separate PMF analyses.**

2. Lines 352 – 373 (SCA factor): m/z 60 has been used to diagnose the secondary combustion aerosol (SCA) factor. If m/z 60 was equivalent to Org60 in Table 4, the analysis for SCA was very confusing. According to the table, Org60 has a lower Pearson correlation coefficient (0.24) with SCA, compared with Org57 (0.36). When the authors mention the weak correlation between SCA and Org 57 in line 355, it is unclear why m/z 60 has a good correlation with SCA in Line 367.

AR: Authors agree that the SCA factor explanation was confusing and efforts to make it more understandable have been made. The paragraph explaining SCA is now as follows:

**The SCA factor had a peak particle size of 44.7 nm at both sites and was interpreted as a secondary aerosol originating from combustion processes (i.e., of liquid fuel such as diesel, oil solid fuel such as biomass and coal, or gas). SCA had relatively weak correlations with the primary traffic emissions (e.g., NO$_x$, BC, CO, m/z 57) data, as could be expected for atmospherically processed aerosol. The strongest Pearson correlation coefficient of 0.56 was observed between SCA and NO$_x$ at the UB site (Table 4). SCA and NO$_x$ at the UB site also had similar diurnal patterns on working days and**

**weekends (Fig. 6). The highest SCA peak was seen approximately 3 hours later than for the TRA1 factor, indicating that the SCA factor included traffic emissions that had been aged/processed a couple of hours in the atmosphere. SCA was found to have an evening peak in addition to the morning rush hour peak (Fig. 4e). The evening peak was more pronounced during weekends, which indicates possible contributions from biomass combustion (Fig. 4f). In an earlier study, BC originating from biomass combustion was shown to contribute 15 ± 14% at the SC and between 41 ± 14 and 46 ± 15% of the BC in residential/detached house areas (Helin et al., 2018). To support this, the diurnal trends of SCA and organic fragments at m/z 60 (Q-ACSM) at the SC were plotted (Fig. 6). The fragments at m/z 60, particularly its fraction of the total OA, have widely been used as a marker for primary wood combustion emissions (Alfarra et al., 2007). The shape of the m/z 60 diurnal profile was similar to the SCA diurnal profile during the workdays and weekends strengthening the assumption of wood combustion contribution to SCA. An important thing to note is that the overall correlation with the m/z 60 was still relatively low (Table 4). The similar rush hour peak of m/z 60 to that of SCA was slightly surprising as the m/z 60 is usually related to biomass combustion and not traffic. The annual variation of SCA is small (Fig. 4g). Likely because although during the wintertime, the amount of biomass combustion increases the amount of sunlight is low, limiting SOA formation, whereas during summer, the amount of biomass burning is lower, but the amount of sunlight increases, thus enhancing SOA formation. In contrast, traffic emissions remain stable throughout the whole year.**

In addition, in the whole text, $ORG_{xx}$ has been changed to m/z xx to make the text more uniform. Additionally, in Tables 3 and 4 the $ORG_{xx}$ form has also been changed to m/z xx. Fig. 6 was also updated as the Fig. 6b was found to have a mistake in the title.

Minor Comments
1. Line 65: What concentrations decrease? Please clarify it.

AR: The sentence is now changed to:

**They stated that in urban environments, the majority of particles are in the size range of 10-100 nm, and the PNC decrease approximately by a factor of 100 when the particle size increases from 100 nm to 1000 nm.**

2. Lines 124 – 128: Please provide the IE values for NO3 and the RIE for NH4 and SO4 in the paper.

AR: The following discussion about the values is now added to the text:

**$IE(NO_3)$ varied over the years and the final correction of the NRPM1 mass was done against the mass concentrations derived from DMPS data as described by Barreira et al., (2021). The RIE for $SO_4$ varied from 0.51-0.61 and for $NH_4$ 3.8-5.32.**

3. Section 2.4: Lots of details about processing data have been covered in the main text. To

improve the readability of the section, please only include the necessary parts in the main text and move the details into the supplement.

AR: The authors realize that a lot of details about processing the data have been given in the article however the data analysis is the novelty of this study and preprocessing the data is important to understand the process therefore authors think that significant amounts of the data can not be moved to supplemental material. However, the author removed some unnecessary repetition from the section.

4. Line 355: Please provide a short description of Org57. This is important for readers who are not familiar with AMS or ACSM measurements.

AR: The m/z 57 is now shortly described in the text:

**Of these m/z 57 ($C_4H_{9+}$, $C_3H_5O+$) is a part of the HOA mass fraction that is linked to traffic exhaust emissions (Crilley et al., 2013; Crippa et al., 2012; Daellenbach et al., 2016; Mohr et al., 2012).**

Also, the new references are added to the reference list:

**Crippa, M., DeCarlo, P. F., Slowik, J. G., Mohr, C., Heringa, M. F., Chirico, R., Poulain, L., Freutel, F., Sciare, J., Cozic, J., Di Marco, C. F., Elsasser, M., Nicolas, J. B., Marchand, N., Abidi, E., Wiedensohler, A., Drewnick, F., Schneider, J., Borrmann, S., Nemitz, E., Zimmermann, R., Jaffrezo, J.-L., Prévôt, A. S. H., and Baltensperger, U.: Wintertime aerosol chemical composition and source apportionment of the organic fraction in the metropolitan area of Paris, Atmos. Chem. Phys., 13, 961–981, https://doi.org/10.5194/acp-13-961-2013, 2013.**

**Mohr, C., DeCarlo, P. F., Heringa, M. F., Chirico, R., Slowik, J. G., Richter, R., Reche, C., Alastuey, A., Querol, X., Seco, R., Peñuelas, J., Jiménez, J. L., Crippa, M., Zimmermann, R., Baltensperger, U., and Prévôt, A. S. H.: Identification and quantification of organic aerosol from cooking and other sources in Barcelona using aerosol mass spectrometer data, Atmos. Chem. Phys., 12, 1649–1665, https://doi.org/10.5194/acp-12-1649-2012, 2012.**

**Crilley, L.R, Ayoko, G.A., Jayaratne, E.R., Salimi, F., Morawska, L.: Aerosol mass spectrometric analysis of the chemical composition of non-refractory PM1 samples from school environments in Brisbane, Australia, Sci. Total Environ., 458–460, 81-89. https://doi.org/10.1016/j.scitotenv.2013.04.007, 201**3.

**Daellenbach, K. R., Bozzetti, C., Křepelová, A., Canonaco, F., Wolf, R., Zotter, P., Fermo, P., Crippa, M., Slowik, J. G., Sosedova, Y., Zhang, Y., Huang, R.-J., Poulain, L., Szidat, S., Baltensperger, U., El Haddad, I., and Prévôt, A. S. H.: Characterization and source apportionment of organic aerosol using offline aerosol mass spectrometry, Atmos. Meas. Tech., 9, 23–39, https://doi.org/10.5194/amt-9-23-2016, 2016.**

5. Line 395: What filters were used in the Niemi et al (2009)? Please clarify.

AR: Now the filter material Teflon is added to the text:

**Additionally, Niemi et al. (2009) did not report high concentrations of NO₃ during the LRT episodes, likely because of evaporation losses of ammonium nitrate from the Teflon filters.**

6. Lines 379 – Lines 386 (SecA factor): Please discuss why the SecA factor has moderate correlation coefficients with Org57 (0.52) and Org60 (0.5) in Table 4.

AR: The following paragraph discussing the possible reason for this is now added to the text:

**In addition, the SecA had moderate correlations with m/z 57 and m/z 60 (Table 4). The reason for this is likely to be the high total amount of organics as the m/z 57 and m/z 60 do not refer to the relative fraction of the total organics but absolute concentrations of the mass fraction and therefore they might be elevated with the higher total organic mass in particles.**

7. Lines 387 – Lines 398 (LRT factor): Please discuss why the LRT factor has considerable correlation coefficients with Org60 (0.64) in Table 4.

AR: The following discussion about the correlation is added to the text:

**The LRT factor had a moderate correlation with m/z 60. This is likely to be caused by the large PM of LRT particles and therefore higher total organic mass. Additionally, the correlation with m/z 60 might indicate the contribution of remote biomass burning to the LRT factor.**

Technical Comment
1. Lines 16 – 17: Please rephrase "… a novel approach to positive matrix factorization (PMF)"

AR: It is now rephrased as follows:

**This study intended to develop the source apportionment of urban aerosols by utilising a novel approach to positive matrix factorization (PMF).**

2. Line 56: The abbreviation for particle number size distribution should be PNSD. Please correct it here, but also other places where NSD are being in use.

AR: OK done.

3. Lines 136 – 137: Provide the manufactures for TEI42S and TEI49.

AR: The manufacturer of the instruments is TEI (Thermo Environmental Instruments) this is now written open in the text.

4. Line 182: Please clarify: "This was also necessary because using too large data files was not possible in EPA PMF 5.0."

AR: This is now clarified in the text:

**This was also necessary because growing the size of data files over a certain point would cause EPA PMF 5.0 program to crash during the analysis because of the program running out of memory.**

5. Line 365: Is m/z 60 the same as the Org60 in Table 4? If so, please make it consistent. Same case for m/z43 in Line 383

AR: This has been done as a part of the answer to major comment 2.

6. Line 430: Is Fig. 8 redundant here?

AR: The Fig. 8 is not redundant here but the sentence is missing other words and it is now corrected to the following form:

**This is in contrast to for example TRA1, which shows little month-to-month variation, and the concentrations stay relatively stable between the different months in Fig. 7 and Fig. 8.**

Reviewer 2# General Comments
It needs to be acknowledged that significant improvement has been made in the revised version of the manuscript. A more thorough introduction is now present, while the authors made an effort to address the comments made by the reviewers, regarding the presentation of the solution selection rationale, as well as regarding the interpretation of the obtained results. Nevertheless, I find it troubling that the authors are still not providing any proof whatsoever that their proposed new methodology is advantageous in figuring out the sources of particles affecting the two sites when compared to other known practices. I would like to stress this and ask the authors to take action to address it. If it is not proven that combining data from two sites in a PMF leading to a common timeseries has a positive effect in the source apportionment results, both from an algorithmic as well as from a physical meaning standpoint, the question about the validity and quality of the results will be overshadowing the whole effort. It is the authors themselves that in lines 429-430 in the "Conclusions and summary" section of the revised manuscript state that "the novel method of attaching simultaneous data from two sites seems to improve the detection of various factors and could be useful in locations where NSD data is available from more than one site", yet no proof of that improvement is provided in the article. Not very helpful in the review process itself is the fact that the authors when replying to my comments from the previous round of reviews, provide unclear and contradicting responses, while in certain occasions they address the comments rather poorly. For instance:

• In their response to General Comment #1, the authors state that "We tested different approaches for the data analysis (with and without additional data, analyzing stations separately, and different numbers of factors)" but a few sentences below they say that "Analyzing this large data set even with one tool was really time-consuming and challenging and thus it was not feasible to use several different approaches". So, what is the case here? Did they or did they not test different approaches?

AR: The authors did test different approaches to the PMF. These ways included doing PMF separately for the two stations using a wide range of factor numbers and also doing PMF with added BC data as columns to input matrixes to PMF. Also, different numbers of factors were tested for horizontally joined PMF. This is reflected in Fig. 2 which shows the mean residuals for factor numbers of 2 to 10. The answer to the first round of reviewer comments: "Analysing this large data set even with one tool was really time-consuming and challenging and thus it was not feasible to use several different approaches" refers to the fact that results from these other methods could not be analysed as thoroughly as the results from horizontally joined PMF 5 factor solution. Meaning that the diurnal profiles, residual analyses and correlations with auxiliary data were not studied for all of the tested approaches to PMF

• When replying to General Comment #2. In the first sentence they state "The authors were not able to do the analysis separately for the stations", and immediately after that they state that "The results were not satisfactory. We tested many different approaches and found this to be the best solution." What I understand from these sentences is that the analysis was done, but the authors deemed it not satisfactory. Proof of that needs to be provided in the revised version.

AR: The authors agree that the answer to General Comment #2 was not very clear. The author's statement "The authors were not able to the analysis separately for the stations. The results were not satisfactory" was intended to mean just that the authors tested doing the PMF analyses separately for the stations but the results were not good enough. The factors were meant to be found in this article using only PNSD data in the PMF input files and only comparing them to the auxiliary data afterwards. By using this approach when the PMF was done separately for the SC and UB, especially at the SC all the factors seemed to be correlating with traffic. This did not seem realistic. Now the main reason why the results were deemed not satisfactory is also stated in the text:

**The decision to join the data together horizontally instead of doing the PMF analysis separately for the SC and UB was made because the idea in this study was to use only PNSD data as input data in PMF analysis and only to use the auxiliary data for identification of the factors. When the PMF analyses were done separately for the stations without additional data, the PMF was able to split the measured PNSD into factors at each site. However, seemingly any number of factors could be fitted, with PMF only fractioning the measured PNSD to more sub-modes. Additionally, when PMF analysis was performed separately the attained factors had different modes and mode concentration in between the stations in all cases. This is not likely to resemble reality as the stations reside less than 1 km from each other and therefore somewhat similar background and long-range transport factors would be expected. Adding the data from the two sites together horizontally forces the PMF to find a common time series between the stations. This is beneficial in finding the common factors for the UB and SC as the time series of the common factors can be expected to be similar because of the small distance between the stations. On the other hand, joining the data horizontally does not force the same factor profiles for both sites. An additional problem of doing the PMF separately for the stations was that all factors at the SC site seemed to correlate strongly with the traffic diurnal cycle indicating that the traffic emissions are split between the different factors. The figures showing the 4, 5, and 6-factor solutions of the PMF analyses done separately for SC and UB are presented in**

**supplemental material S3, S4, and S5 respectively. The negative side of merging the data horizontally can be expected to lower the total explained a fraction of the PNSD while the common time series are forced for both of the stations.**

The following figures are added to the supplement:

[Figure]

**Figure S3: Four-factor solutions, contributions during workdays, weekends, and different months from PMF analyses done separately for SC (left column) and UB (right column).**

[Figure]

**Figure 4S: Five-factor solutions, contributions during workdays, weekends, and different months from PMF analyses done separately for SC (left column) and UB (right column).**

[Figure]

**Figure S5: Six-factor solutions, contributions during workdays, weekends, and different months from PMF analyses done separately for SC (left column) and UB (right column).**

• Regarding the authors response to General Comment #3. I fully understand the challenges surrounding the process of introducing, and for the first time applying a new methodology. I guess it requires thorough knowledge and understanding of existing approaches in the authors part, but more importantly, the proposed method needs to be able to perform equally or outperform existing approaches, if not in the general case, at least when applied in a specific dataset. After the new method's performance is presented, then the reader can really focus on the method's results and their significance in understanding the atmospheric processes under investigation. In the form that the manuscript is now, the reader is left uninformed on whether this proposed way of treating PNSD data from different types of

stations is worth the effort. I genuinely think that horizontally combining the datasets is a brilliant idea. If it really works similarly or even better to single station PMF then it could be recommended to everybody interested in contrasts between different types of stations. I would like to urge the authors to perform single station PMF tests and present their comparison to the combined PMF here. They could do it in a smaller but still representative portion of the dataset of their choosing, if they feel that a full comparison is time consuming. If the authors are still reluctant on performing such tests, at least they could perform a comparison with the overlapping years presented in Rivas et al. (2020). In any case, I believe that any advancements and/or limitations related to the new methodology need to be discussed in the text.

AR: The PMF analyses were redone for both stations separately for factor numbers of 4-6. The results from these comparisons are now presented in supplemental material as stated in the author's response to the previous comment of the reviewer. The discussion about the benefits and drawbacks of the method are now discussed in the text. This modified text has been attached to this document as a response to the previous comment of the reviewer.

• With General Comment #4, along with several comments from Reviewer 1, the authors were given the chance to provide sufficient information on the PMF quality. They partly addressed the matter, but still some features can be added. The following should be added in the supplementary material:

AR: Answering some of these questions afterwards is a slightly challenging as EPA PMF only gives 5 text files as an output after the analysis is done and some information is lost when the program is turned off. These files include residual matrixes, contributions of factors at each given time, diagnostics, source profiles and base run comparisons.

o a comparison (regression) between the total (Ntot) particle number concentration (PNC) measured at each site and the respective PMF reconstructed PNC.

AR: This is now done, and the figures have been added to the supplements:

[Figure]

**Figure 9S: Regression plots between modelled and measured interpolated concentrations at SC (A) and UB (B).**

The following text was also added to the manuscript:

**Also, the figures showing the regressions between the modelled and measured concentrations after interpolation are presented in supplemental material S9.**

o what was the unexplained fraction of Ntot? Any temporal trends there?

AR: The average unexplained fraction for SC was 12.7 % & and 6.7 % for UB. There were slight temporal trends with the unexplained factor increasing over the daytime hours at SC. In the monthly plot, no clear trend was observed. The unexplained percentages are now presented in the text:

**The average difference between these was 12.7 % at SC and 6.7 % at UB and the temporal patterns for the difference have been presented in supplemental material S10.**

Additionally, a figure showing the diurnal variation of the unexplained parts during workdays and weekends with monthly unexplained fractions is presented in the supplement:

[Figure]

**Figure S10: Average differences between modelled and measured concentrations for workdays (A), weekends (B) and months (C)at SC and UB.**

*o a short description of the DISP process results (sawps, dQ percentage etc)*

AR: The DISP process was run by PMF and there were no swaps or dQ drops. This is also now mentioned in the text:

**The results of the displacement analysis showed no drop in Q values or swaps in any of the analyses.**

o how many random seed runs were performed for the presented solution?

AR: The number of seed runs used was 5. Using a larger number caused EPA PMF 5.0 to crash. However, The analysis was run several times and the results were similar at all of them. The number of used seeds is now also mentioned in the text:

**The robustness of the solution was tested using five different random seeds as starting points. Performing analysis with a larger seed number sometimes caused the program to crash. In addition to using random sees a displacement analysis was performed on the solutions.**

o was there any variability between each run's factor contribution to Ntot?

AR: The data from the other seed runs is not saved when using EPA PMF 5.0 so I can not calculate this. However, the output files of PMF files show that all the runs had the same exact Qrobust values and only minimal differences in the Qtrue value indicating that the differences between the results of different runs were minimal.

o were there any tools used to address rotational ambiguity (e.g. fpeak)? I would assume that there were, so please provide the appropriate details (G-space plots etc).

AR: The rotational ambiguity was tested only based on displacement analysis. The results of displacement analysis demonstrated no rotational ambiguity so this was not further tested.

• Regarding the authors response to General Comment #6. Given that the authors have performed wind regression to the PMF results, these results should be presented at least in the supplementary material. What do the authors mean when saying that this type of analysis did not provide any additional information? Didn't the polar plots of each factor point to different contributions for different wind directions and speeds? About the SC site, I'm not sure I fully agree. For instance, Rivas et al. (2020) performed the analysis and presented it, providing some insightful information.

AR: The authors have studied the effect of wind directions on the source factor contributions. This has been done by drawing wind roses for contributions of factors from different wind directions and it is now added to the supplemental material and text referring to the figure added to the text:

**These contributions were also calculated for each factor separately in supplemental material S22 for different wind directions.**

The following figure was also added to the supplemental material:

[Figure]

**Figure S22: Scaled contributions of LRT, SCA, SecA, TRA1 and TRA2 factors from different wind directions.**

---

## Author Response (AR3)

We thank the reviewer for taking the time to read the article and for valuable comments on our paper. To facilitate the revision process, we have copied the reviewer comments (in black text) and our responses are in blue font. We have responded to all the reviewer comments and made alterations to our paper (**in bold text**).

**Editor:**

**Public justification (visible to the public if the article is accepted and published)**:
The main problem identified is now addressed with the new paragraph at the end of section 2.4. The other minor points have been largely addressed and while I would venture that some interpretations may remain contentious, this is somewhat inevitable with new methodologies, so I do not think publication needs to be held up further in terms of scientific merit.

The following technical needs fixing before final publication:
Table 1: Simply describing the instrument with a number after the deletion of "TEI" is not sufficient; it would be better to say something like "Thermo Environmental model 49" or similar. I would generally harmonise the entire column, because it is currently a mixture of ordering of instrument description, manufacturer and model number, with different layouts on different rows.

The column presenting the instruments in Table 1 is now harmonised and one related reference is added to the reference list:

**Table 1: The list of instruments used in the measurements.**

| Instrument | Station | Measured variable |
|---|---|---|
| DMPS (Airmodus CPC A20 with Vienna type DMA) | SC | PNSD |
| DMPS (Twin-DMPS, Aalto et al. 2001) | UB | PNSD |
| Q-ACSM (Aerodyne Research) | SC | Non-refractory $PM_1$ |
| TEOM (model 1405, Thermo Scientific) | SC | $PM_{10}$ and $PM_{2.5}$ |
| Aethalometer (AE33, Magee Scientific) | SC | BC |
| Ambient $NO_x$ Monitor (APNA-370, Horiba) | SC | $NO_x$ |
| Ambient Ozone Monitor (APOA-370, Horiba) | SC | $O_3$ |
| LICOR (model LI-7000) | SC | $CO_2$ |
| Ambient Carbon Monoxide Monitor (APMA-360, Horiba) | SC | CO |
| $NO$-$NO_2$-$NO_x$ Analyzer (42C, Thermo Environmental Instruments) | UB | $NO_x$ |
| $O_3$ Analyzer (model 49C, Thermo Environmental Instruments) | UB | $O_3$ |

| | | |
|---|---|---|
| Ambient Carbon Monoxide Monitor (APMA- 370, Horiba) | UB | CO |
| Ambient Sulfur Dioxide Analyzer (APSA -370, Horiba) | UB | $SO_2$ |

**Aalto, P., Hämeri, K., Becker, E. D. O., Weber, R., Salm, J., Mäkelä, J. M., Hoell, C., O'Dowd, C. D., Karlsson, H., Hansson, H-C., Väkevä, M., Koponen, I. K., Buzorius, G., Kulmala, M., Physical characterization of aerosol particles during nucleation events, Tellus B, 53(4), 344-358, https://doi.org/10.1034/j.1600-0889.2001.530403.x, 2001.**

Additionally, also one typo was corrected in the reference list and unnecessary piece of text was removed from Table 3 after the term TRA2.